# LVM-Med: Learning Large-Scale Self-Supervised Vision Models for Medical Imaging via Second-order Graph Matching

**Duy M. H. Nguyen**[*1,2,3], **Hoang Nguyen**[3], **Nghiem T. Diep**[3], **Tan N. Pham**[3,4], **Tri Cao**[3],
**Binh T. Nguyen**[4], **Paul Swoboda**[5], **Nhat Ho**[6], **Shadi Albarqouni**[7,8], **Pengtao Xie**[9,10],
**Daniel Sonntag**[†3,11], **Mathias Niepert**[*†1,2]

[1]University of Stuttgart, [2]IMPRS for Intelligent Systems
[3]German Research Center for Artificial Intelligence, [4]University of Science - VNUHCM,
[5]Max Planck Institute for Informatics, [6]University of Texas at Austin [7]Helmholtz Munich,
[8]University Hospital Bonn, [9]UC San Diego, [10]MBZUAI, [11]Oldenburg University.

## Abstract

Obtaining large pre-trained models that can be fine-tuned to new tasks with limited annotated samples has remained an open challenge for medical imaging data. While pre-trained deep networks on ImageNet and vision-language foundation models trained on web-scale data are prevailing approaches, their effectiveness on medical tasks is limited due to the significant domain shift between natural and medical images. To bridge this gap, we introduce LVM-Med, the first family of deep networks trained on large-scale medical datasets. We have collected approximately 1.3 million in medical images from 55 publicly available datasets, covering a large number of organs and modalities such as CT, MRI, X-ray, and Ultrasound. We benchmark several state-of-the-art self-supervised algorithms on this dataset and propose a *novel self-supervised contrastive learning algorithm using a graph matching formulation*. The proposed approach makes three contributions: (i) it integrates prior pair-wise image similarity metrics based on local and global information; (ii) it captures the structural constraints of feature embeddings through a loss function constructed via a combinatorial graph-matching objective; and (iii) it can be trained efficiently end-to-end using modern gradient-estimation techniques for black-box solvers. We thoroughly evaluate the proposed LVM-Med on 15 downstream medical tasks ranging from segmentation and classification to object detection, and both for the in and out-of-distribution settings. LVM-Med empirically outperforms a number of state-of-the-art supervised, self-supervised, and foundation models. For challenging tasks such as Brain Tumor Classification or Diabetic Retinopathy Grading, LVM-Med improves previous vision-language models trained on 1 billion masks by 6-7% while using only a ResNet-50. We release pre-trained models at this link `https://github.com/duyhominhnguyen/LVM-Med`.

## 1  Introduction

Constructing large-scale annotated medical image datasets for training deep networks is challenging due to data acquisition complexities, high annotation costs, and privacy concerns [1, 2]. Vision-language pretraining has emerged as a promising approach for developing foundational models that support various AI tasks. Methods such as CLIP [3], Align [4], and Flava [5] propose a unified model trained on large-scale image-text data, showing exceptional capabilities and performance across

---

[*]Corresponding authors, [†]Co-Senior authors.

37th Conference on Neural Information Processing Systems (NeurIPS 2023), New Orleans.

various tasks. However, their effectiveness in the medical domain still remains unclear. A recent work SAM [6] trains large vision models on over one billion annotated masks from 11M natural images, enabling interactive segmentation. Nevertheless, SAM's zero-shot learning performance is moderate on other datasets [7, 8], highlighting the need for fine-tuning to achieve satisfactory results [9].

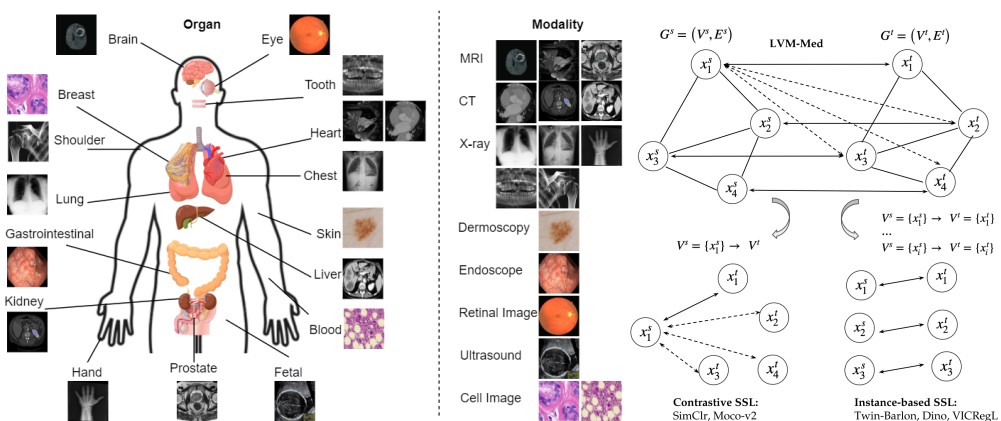

Figure 1: (left) Overview of the body organs and modalities in our collected dataset; (right) LVM-Med unifies and extends contrastive and instance-based self-supervised learning approaches by specifying graph's properties.

To facilitate the development of foundation models in the medical domain, we make two major contributions. First, we have curated a vast collection of 55 publicly available datasets, resulting in approximately 1.3 million medical images covering various body organs and modalities such as CT, MRI, X-ray, ultrasound, and dermoscopy, to name a few. Second, we propose LVM-Med, a novel class of contrastive learning methods, utilizes pre-trained ResNet-50 and a ViT network SAM[10]. We evaluate various instances of LVM-Med relative to popular supervised architectures and vision-language models across 15 medical tasks. To our best knowledge, this is the first time such a large-scale medical dataset has been constructed and used to investigate the capabilities of SSL algorithms.

LVM-Med incorporates a second-order graph-matching formulation, which subsumes and extends a large class of contrastive SSL methods. Given a batch of images, two random transformations are applied to each image, and the resulting transformed images are then fed to an image encoder. The embedding vectors obtained from images in a batch are used to construct two graphs where vertices represent pairs of transformed images generated from the same original one. Through solving a graph-matching problem [11, 12], we learn feature representation such that their encoding serves as suitable priors for a global solution of the graph-matching objective. This approach is distinct from prior contrastive learning methods that focus on merely optimizing pair-wise distances [13, 14] between transformed images or learning contrastive distances with positive and negative samples [15–19]. It is worthwhile noting that previous contrastive learning methods are special instances of our general framework (Figure (1), right).

LVM-Med has several advantages over existing approaches. First, it integrates advanced pair-wise image similarity taken from prior SSL methods into vertex affinities, resulting in both global and local information that can be efficiently fused. Second, it uncovers underlying structures of feature embeddings by utilizing edge constraints, enhancing robustness in the presence of similar entities in medical datasets. Third, though combinatorial problems are typically non-differentiable, LVM-Med can efficiently calculate gradients through the discrete combinatorial loss function using modern implicit maximum likelihood estimation techniques. Consequently, LVM-Med can scale successfully on large-scale data. In a wide range of 15 medical experiments, LVM-Med sets a new state-of-the-art in fully fine-tuning or prompt-based segmentation, linear and fully fine-tuning image classification, and domain generalization, outperforming several vision-language models trained on a hundred million image-text instances.

We summarize major contributions in this work, including:

(i) We present a collection of large-scale medical datasets, serving as a resource for exploring and evaluating self-supervised algorithms.

(ii) We propose LVM-Med, a novel SSL approach based on second-order graph matching. The proposed method is flexible in terms of integrating advanced pair-wise image distance and being able to capture structural feature embedding through the effective utilization of second-order constraints within a global optimization framework.

(iii) On both ResNet-50 and ViT architectures, LVM-Med consistently outperforms multiple existing self-supervised learning techniques and foundation models across a wide range of downstream tasks.

## 2   Related Work

### 2.1   Self-supervised learning in medical image analysis

The latest approaches of *global feature* SSL rely on shared embedding architecture representations that remain invariant to different viewpoints. The variation lies in how these methods prevent collapsing solutions. *Clustering methods* [20–22] constrain a balanced partition of the samples within a set of cluster assignments. *Contrastive methods* [15–18] uses negative samples to push far away dissimilar samples from each other through contrastive loss, which can be constructed through memory bank [23], momentum encoder [24], or graph neural network [19]. Unlike contrastive learning, *instant-based learning* depends on maintaining the informational context of the feature representations by either explicit regularization [13, 25] or architectural design [26, 27]. Our work relates to contrastive and instance-based learning, where a simplified graph-matching version of 1-N or 1-1 reverts to these approaches.

In contrast to global methods, *local methods* specifically concentrate on acquiring a collection of local features that depict small portions of an image. A contrastive loss function can be used on those feature patches at different criteria such as image region levels [28], or feature maps [29, 14]. These strategies are also widely applied in the medical context, thereby pre-text tasks based on 3D volume's properties, such as reconstructing the spatial context [30], random permutation prediction [31] and self-restoration [32, 33], are proposed. Our LVM-Med model on this aspect can flexible unifying both global and local information by adding them to the affinities matrixes representing the proximity of two graphs, enhancing expressive feature representations.

### 2.2   Vision-language foundation models

In order to comprehend the multi-modal world using machines, it is necessary to create foundational models that can operate across diverse modalities and domains [34]. CLIP [3] and ALIGN [4] are recognized as groundbreaking explorations in foundation model development. These models demonstrate exceptional proficiency in tasks such as cross-modal alignment and zero-shot classification by learning contrastive pretraining on extensive image-text pairs from the web, despite the presence of noise. To further support multi-modal generation tasks such as visual question answering or video captioning, recent works such as FLAVA [5] and OmniVL [35] are designed to learn cross-modal alignment as well as image-video language models. Conversely, the SAM model [6] utilized a supervised learning strategy with over 1 billion masks on 11 million user-prompt interactions and achieved impressive zero-shot segmentation performance on unseen images. While many efforts have been proposed for natural image domains, limited research has been conducted on large-scale vision models for medical imaging. This motivated us to develop the LVM-Med model.

### 2.3   Graph matching in visual computing

Graph matching is a fundamental problem in computer vision, which aims to find correspondences between elements of two discrete sets, such as key points in images or vertices of 3D meshes, and used in numerous vision tasks, including 3D vision [36, 37], tracking [38, 39], shape model learning [40, 41], and many others [42–45]. In this framework, the vertices of the matched graphs correspond to the elements of the discrete sets to be matched. Graph edges define the cost structure of the problem, namely, second order, where pairs of matched vertices are penalized in addition to the vertex-to-vertex matchings. This allows us to integrate the underlying geometrical relationship between vertices into account but also makes the optimization problem NP-hard. Therefore, many approximate approaches have been proposed to seek acceptable suboptimal solutions by relaxing discrete constraints [46, 47]. In other directions, gradient estimation techniques for black-box solvers are employed to make the hybrid discrete-continuous matching framework be differentially end-to-end [48–50]. Our LVM-Med follows the latter direction and, for the first time, presents the formulation of contrastive learning as a second-order graph-matching problem.

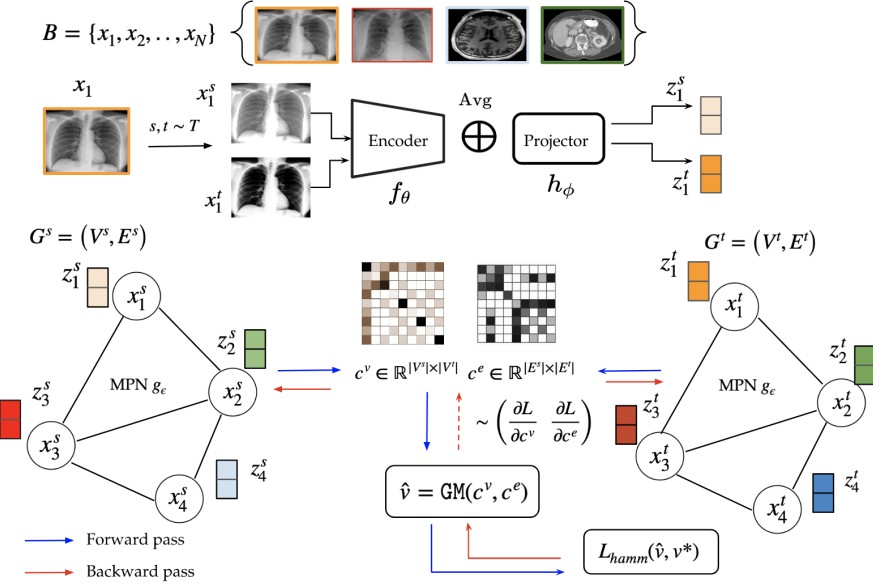

Figure 2: LVM-Med Overview. $\mathrm{Avg}$ is the average pooling layer, MPN denotes for message passing network, $\mathrm{GM}$ indicates the combinatorial solver, and $(c^v, c^e)$ represents vertex and edge affinity matrices. For each image $\boldsymbol{x}_i$ in batch size, we generated two distorted versions and fed them into the feature representation $f_\theta$ and another projector $h_\theta$. The obtained embeddings $\boldsymbol{z}_i^\ell$, $\ell \in (s, t)$ are used to build two graphs $G^s, G^t$. We further design a message passing network $g_\epsilon$ that aggregate feature per node by their neighbor information. Then we compute vertex and edge affinities $\boldsymbol{c}^v, \boldsymbol{c}^e$ and use them to solve the graph matching. The output afterward is compared with pairs of ground truth $(\boldsymbol{x}_i^s, \boldsymbol{x}_i^t)$, $i \in (1, .., N)$ representing distorted images generated from the same sample. In the backward pass, we use modern gradient-estimation techniques to approximate $\frac{\partial L}{\partial \boldsymbol{c}^v}$ and $\frac{\partial L}{\partial \boldsymbol{c}^e}$ .

# 3 Methodology

## 3.1 Dataset construction

We provide detailed information about the collected datasets in the Appendix. The data was collected from publicly available resources, which include a diverse set of modalities and body organs as illustrated in Figure 1 (left). The data format is a combination of 2D images and 3D volumes as well as X-ray, MRI, CT, Ultrasounds, etc. For datasets whose data dimensions are 3D volumes, we slice them into 2D images. To avoid potential test data leaking for downstream tasks, we use the default training partition in each dataset; otherwise, we randomly sample with $20\%$ total images. In total, we obtain approximately $1.3$ million images. More statistics on the dataset are presented in the Appendix.

## 3.2 Contrastive learning as graph matching

Figure 2 provides an illustration of our LVM-Med method, which learns the feature representation $f_\theta$ by matching two distorted views derived from the same input image through a graph-matching formulation. Below we describe in detail each component.

### 3.2.1 Graph construction on feature embedding

Given a batch of $N$ images $\boldsymbol{B} = \{\boldsymbol{x}_1, \boldsymbol{x}_2, .., \boldsymbol{x}_N\}$ sampled from a dataset, we generate for each image $\boldsymbol{x}_i \in \boldsymbol{B}$ two transformed images $\boldsymbol{x}_i^s$ and $\boldsymbol{x}_i^t$ by using two transformations $s, t \sim T$ sampled from $T$, a set of pre-defined image transformations. After the transformations, each image is of shape $(C \times H \times W)$, where $C$ is the number of channels and $(H, W)$ the original spatial dimensions. These distorted images are fed into an encoder $f_\theta : \mathbb{R}^{C \times H \times W} \to \mathbb{R}^{D \times R \times S}$ to produce two representations $\boldsymbol{y}_i^s = f_\theta(\boldsymbol{x}_i^s)$ and $\boldsymbol{y}_i^t = f_\theta(\boldsymbol{x}_i^t)$ where $D$ is the number of feature channels and $(R, S)$ are the spatial dimensions of the feature map. On each such representation, we perform an average pooling

operation $\text{Avg} : \mathbb{R}^{D \times R \times S} \to \mathbb{R}^D$ followed by another projection $h_\phi : \mathbb{R}^D \to \mathbb{R}^F$ to form two feature embeddings $\boldsymbol{z}_i^s = h_\phi(\text{Avg}(\boldsymbol{y}_i^s))$, and $\boldsymbol{z}_i^t = h_\phi(\text{Avg}(\boldsymbol{y}_i^t)) \in \mathbb{R}^F$ with $F < D$.

Given a set of embeddings for a batch $\boldsymbol{B}$, we construct two graphs $G^s$ and $G^t$ where, for each pair $(\boldsymbol{x}_i^s, \boldsymbol{x}_i^t)$ of corresponding distorted images, we add a node representing $\boldsymbol{x}_i^s$ to $G^s$ and a node representing $\boldsymbol{x}_i^t$ to $G^t$. Hence, for each $\ell \in \{s, t\}$, we construct a graph $G^\ell = (V^\ell, E^\ell)$ with $V^\ell = \{\boldsymbol{x}_1^\ell, ..., \boldsymbol{x}_N^\ell\}$ the set of vertices and $E^\ell$ the set of edges $e_{ij}^\ell = (\boldsymbol{x}_i^\ell, \boldsymbol{x}_j^\ell)$. The node-level feature matrix is given by $\boldsymbol{X}^\ell = \left[ \boldsymbol{z}_1^\ell; ...; \boldsymbol{z}_N^\ell \right] \in \mathbb{R}^{N \times F}$ which associates each vertex $\boldsymbol{x}_i^\ell$ with its feature embedding $\boldsymbol{z}_i^\ell$. We create edges for each graph $G^\ell$ through a $k$-nearest neighbors algorithm using the feature matrix $\boldsymbol{X}^\ell$. The adjacency matrix $\boldsymbol{A}^\ell \in \mathbb{R}^{N \times N}$ is defined as $A_{ij}^\ell = 1$ if $e_{ij}^\ell \in E^\ell$ and $A_{ij} = 0$ otherwise. With the two graph structures given, we obtain a node-attributed graph $G^\ell = (V^\ell, \boldsymbol{A}^\ell, \boldsymbol{X}^\ell)$ on which a graph neural network $g_\varepsilon$ is used to aggregate the nodes' features. In particular, $g_\varepsilon$ computes an embedding $\hat{\boldsymbol{Z}}^\ell = g_\varepsilon(\boldsymbol{X}^\ell, \boldsymbol{A}^\ell)$ by performing message passing operations. We set $g_\varepsilon$ to be a graph convolutional network [51, 52] consisting of $l+1$ layers $g_\varepsilon = \{g_l, g_{l-1}, .., g_0\}$ where the output of layer $l$ is computed as

$$H_l^\ell = \sigma \left( \tilde{D}^{-\frac{1}{2}} (\boldsymbol{A}^\ell + \boldsymbol{I}_N) \tilde{D}^{-\frac{1}{2}} H_{l-1}^\ell g_{l-1} \right), \tag{1}$$

where $\boldsymbol{I}_N$ is the identity matrix modeling self-connections; $\tilde{D}$ is a diagonal matrix with $\tilde{D}_{ii} = \sum_j \boldsymbol{A}_{ij}^\ell$; $g^{l-1}$ are the trainable parameters for each layer; $\sigma(\cdot)$ is an activation function; and $H_0^\ell = \boldsymbol{X}^\ell$. We use the outputs of the last layer as embeddings for the nodes, that is, $\hat{\boldsymbol{Z}}^\ell = H_l^\ell \in \mathbb{R}^{N \times F}$ given the shared graph network $g_\varepsilon$.

We now have two graphs $G^s, G^t$ with node attribute matrices $\hat{\boldsymbol{Z}}^s, \hat{\boldsymbol{Z}}^t$, the outputs of the graph neural networks. Next, a graph-matching problem is constructed and solved where the gold matching is given by the pairs $(\boldsymbol{x}_i^s, \boldsymbol{x}_i^t) \ \forall i \in \{1, .., N\}$.

### 3.2.2 Learning affinities with global and local context

To represent potential connections for a pair of node $(\boldsymbol{x}_i^s, \boldsymbol{x}_a^t)$ where $\boldsymbol{x}_i^s \in G^s, \ \boldsymbol{x}_a^t \in G^t$, we design a vertex affinity matrix $\boldsymbol{c}^v \in \mathbb{R}^{|V^s||V^t|}$ where $c_{ia}^v$ is the prior (feature-based) similarity between $\boldsymbol{x}_i^s$ and $\boldsymbol{x}_a^t$. An advantage of our formulation is its ability to integrate advanced pair-wise distance can be smoothly integrated to $c_{ia}^v$, resulting in more expressive proximity representation. In particular, we leverage both global and local consistency derived from feature embeddings of distorted images. The *global distance* used in several prior works can be computed as $c_{ia}^{glo}(\boldsymbol{x}_i^s, \boldsymbol{x}_a^t) = \cos(\hat{\boldsymbol{z}}_i^s, \hat{\boldsymbol{z}}_a^t)$ where $\cos(\cdot)$ denotes cosine similarity; $\hat{\boldsymbol{z}}_m^\ell$ is the embedding of $\boldsymbol{x}_m^\ell$ ($\ell \in \{s, t\}, \ m \in \{i, a\}$) obtained after message passing in Eq. (1).

Compared to global methods that implicitly learn features for the entire image, local methods concentrate on explicitly learning a specific group of features that characterize small regions of the image. As a result, they are more effective for dense prediction tasks such as segmentation [29, 14, 53]. While recent works applied these tactics as a part of pair-wise minimization conditions [54, 28] Instead, we integrate them as a part of vertex costs $c_{ia}^v$ and use it to solve the graph matching problem. Indeed, we adapt both location- and feature-based local affinity computed as:

$$c_{ia}^{lo}(\boldsymbol{x}_i^s, \boldsymbol{x}_a^t) = \mathbb{E}_{p \in \boldsymbol{P}} \cos(\boldsymbol{q}_p^s, \boldsymbol{q}_{\text{m}(p)}^t) + \mathbb{E}_{p \in \boldsymbol{P}} \cos(\boldsymbol{q}_p^s, \boldsymbol{q}_{\text{m}'(p)}^t) \tag{2}$$

where $\boldsymbol{P} = \{(r, s) | (r, s) \in [1, ..., R] \times [1, .., S]\}$ be the set of coordinates in the feature map $\boldsymbol{y}_i^s \in \mathbb{R}^{D \times R \times S}$ of $\boldsymbol{x}_i^s$; $\boldsymbol{q}_p^\ell$ ($\ell \in \{s, t\}$) be the feature vector at position $p$; $\text{m}(p)$ denote the spatial closest coordinate to $p$ in coordinates of feature map $\boldsymbol{y}_a^t$ estimated through transformations on original image $\boldsymbol{x}_i$; finally $\text{m}'(p)$ represents the closest feature vector to $p$ in $\boldsymbol{y}_a^t$ using $l^2$ distance. Intuitively, the local cost in Eq. (2) enforces invariance on both spatial location and between embedding space at a local scale. Our final affinity cost is computed as:

$$c_{ia}^v(\boldsymbol{x}_i^s, \boldsymbol{x}_a^t) = \alpha \left( c_{ia}^{glo}(\boldsymbol{x}_i^s, \boldsymbol{x}_a^t) \right) + (1 - \alpha) \left( c_{ia}^{lo}(\boldsymbol{x}_i^s, \boldsymbol{x}_a^t) + c_{ia}^{lo}(\boldsymbol{x}_a^t, \boldsymbol{x}_i^s) \right) \tag{3}$$

### 3.2.3 Self-supervision through second-order graph matching

While the standard graph matching problem for vertex-to-vertex correspondences can be used in our setting (LAP), it fails to capture the similarity between edges. If there are duplicated entities

represented by distinct nodes in the same graph, the LAP will consider them identical and skip their neighboring relations. For instance, during the image sampling, two consecutive image slides were sampled from a 3D volume, resulting in their appearances have s a small difference. In such cases, it is complicated to correctly identify those augmented images generated from the same one without using information from the relations among connected nodes in the constructed graph. To address this problem, we introduce additional edge costs $\boldsymbol{c}^e \in \mathbb{R}^{|E^s||E^t|}$ where $c^e_{ia,jb}$ represents the similarity between an edge $v^s_{ij} = \left(\boldsymbol{x}^s_i, \boldsymbol{x}^s_j\right) \in E^s$ and $v^t_{ab} = \left(\boldsymbol{x}^t_a, \boldsymbol{x}^t_b\right) \in E^t$. These edge costs (second-order) are computed as $c^e_{ia,jb} = \cos((\hat{\boldsymbol{z}}^s_i - \hat{\boldsymbol{z}}^s_j), (\hat{\boldsymbol{z}}^t_a - \hat{\boldsymbol{z}}^t_b))$.

We now establish the second-order graph-matching problem. Denoting $\boldsymbol{v} = \{0,1\}^{|V^s||V^t|}$ be indicator vector of matched vertices, i.e., $v_{ia} = 1$ if the vertex $\boldsymbol{x}^s_i \in V^s$ is matched with $\boldsymbol{x}^t_a \in V^t$ and $v_{ia} = 0$ otherwise. The node correspondence between two graphs $G^s$ and $G^t$ that minimizes the global condition stated as:

$$\mathtt{GM}(\boldsymbol{c}^v, \boldsymbol{c}^e) = \arg\min_{\boldsymbol{v} \in U(\mathbf{1}, \mathbf{1})} - \sum_{i,a} c^v_{ia} v_{ia} - \sum_{i,j,a,b} c^e_{ia,jb} v_{ia} v_{jb}$$
$$\text{where} \quad U(\mathbf{1}, \mathbf{1}) = \{\boldsymbol{v} \in \{0,1\}^{N \times N} | \boldsymbol{v}\mathbf{1}_N = \mathbf{1}, \boldsymbol{v}^T \mathbf{1}_N = \mathbf{1}\} \tag{4}$$

and $\mathbf{1}_N$ be a $n$-dimension one-value vector. The constraint $U(\mathbf{1}, \mathbf{1})$ restricts $\boldsymbol{v}$ satisfying the one-to-one matching. Essentially, the Eq. (4) solves the vertex-to-vertex correspondence problem using both node and edges affinities, which can be seen as a form of structural matching (Figure (1),right) and generally can be integrated with higher-order graph constraints as triangle connections or circles. In the experiment, we found out that Eq. (4) significantly improved downstream task performance compared to the pure linear matching approach (Table (6)). Since the Eq. 4 in general is an NP-Hard problem [55] due to its combinatorial nature, we thus use efficient heuristic solvers based on Lagrange decomposition techniques [56].

### 3.2.4 Backpropagating through a graph matching formulation

With $\hat{\boldsymbol{v}} = \mathtt{GM}(\boldsymbol{c}^v, \boldsymbol{c}^e)$ a solution obtained from the solver, we use the Hamming distance and an optimal solution $\boldsymbol{v}^*$ to define the following loss function

$$L(\hat{\boldsymbol{v}}, \boldsymbol{v}^*) = \hat{\boldsymbol{v}}.(1 - \boldsymbol{v}^*) + \boldsymbol{v}^*.(1 - \hat{\boldsymbol{v}}). \tag{5}$$

The proposed approach aims to learn the feature representation function $f_\theta$ such that its output minimizes Eq. (5). However, this is a difficult problem because the partial derivatives of the loss function w.r.t vector costs $\boldsymbol{c}^v, \boldsymbol{c}^e$, i.e., $\partial L/\partial \boldsymbol{c}^v$ and $\partial L/\boldsymbol{c}^e$, are zero almost everywhere [48, 57] due to the objective function in Eq. (4) being piece-wise constant, preventing direct gradient-based optimization.

To approximate the gradients required for backpropagation, we adopt IMLE [50, 58]. Let $\boldsymbol{\theta} = (\boldsymbol{c}^v, \boldsymbol{c}^e)$ be the input to the combinatorial graph matching problem in Eq. (4). The core idea of IMLE is to define a probability distribution $\rho(\boldsymbol{v}; \boldsymbol{\theta})$ over solutions of the combinatorial optimization problem, where the probability of a solution is proportional to its negative cost, and to estimate $\partial L/\partial \boldsymbol{\theta}$ through the gradients of the expectation $\nabla_{\boldsymbol{\theta}} \mathbb{E}_{\hat{\boldsymbol{v}} \sim \rho(\boldsymbol{v}; \boldsymbol{\theta})} [L(\hat{\boldsymbol{v}}, \boldsymbol{v}^*)]$. Since exact sampling from $\rho(\boldsymbol{v}; \boldsymbol{\theta})$ is typically intractable, IMLE instead chooses a noise distribution $\rho(\boldsymbol{\epsilon})$ and approximates the gradient of the expectation over $\rho(\boldsymbol{v}; \boldsymbol{\theta})$ with the gradient of the expectation over $\rho(\boldsymbol{\epsilon})$

$$\nabla_{\boldsymbol{\theta}} \mathbb{E}_{\hat{\boldsymbol{v}} \sim \rho(\boldsymbol{v}; \boldsymbol{\theta})} [L(\hat{\boldsymbol{v}}, \boldsymbol{v}^*)] \approx \nabla_{\boldsymbol{\theta}} \mathbb{E}_{\boldsymbol{\epsilon} \sim \rho(\boldsymbol{\epsilon})} [L(\mathtt{GM}(\boldsymbol{\theta} + \boldsymbol{\epsilon}), \boldsymbol{v}^*)].$$

The above approximation invokes the reparameterization trick for a complex discrete distribution. A typical choice for $\rho(\boldsymbol{\epsilon})$ is the Gumbel distribution, that is, $\rho(\boldsymbol{\epsilon}) \sim \mathrm{Gumbel}(0, 1)$ [59]. Now, by using a finite-difference approximation of the derivative in the direction of the gradient of the loss $\nabla_{\tilde{\boldsymbol{v}}} L(\tilde{\boldsymbol{v}}, \boldsymbol{v}^*)$, we obtain the following estimation rule:

$$\nabla_{\boldsymbol{\theta}} \mathbb{E}_{\hat{\boldsymbol{v}} \sim p(\boldsymbol{v}; \boldsymbol{\theta})} [L(\hat{\boldsymbol{v}}, \boldsymbol{v}^*)] \approx \mathbb{E}_{\boldsymbol{\epsilon} \sim \rho(\boldsymbol{\epsilon})} \left[\frac{1}{\lambda} \left\{ \tilde{\boldsymbol{v}} - \mathtt{GM}\left(\boldsymbol{\theta} + \boldsymbol{\epsilon} - \lambda \nabla_{\tilde{\boldsymbol{v}}} L(\tilde{\boldsymbol{v}}, \boldsymbol{v}^*)\right) \right\} \right], \tag{6}$$

**Algorithm 1** Forward and Backward Pass for $c^v, c^e$

---

**function** FORWARDPASS($c^v, c^e$)
  *// Gumbel noise distribution sampling*
  $\epsilon, \epsilon' \sim \text{Gumbel}(0, 1)$
  *// Graph-matching with perturbed $(c^v, c^e)$*
  $\tilde{v} = \text{GM}(c^v + \epsilon, c^e + \epsilon')$
  *// Save values for the backward pass*
  **save** $(c^v, c^e), (\epsilon, \epsilon')$ and $\tilde{v}$
  **return** $\tilde{v}$

**function** BACKWARDPASS($\nabla_{\tilde{v}} L(\tilde{v}, v^*), \lambda$)
  **load** $(c^v, c^e), (\epsilon, \epsilon')$ and $\tilde{v}$
  *// Add gradient-based pertubations*
  $(c_\lambda^v, c_\lambda^e) = (c^v + \epsilon, c^e + \epsilon') - \lambda \nabla_{\tilde{v}} L(\tilde{v}, v^*)$
  *// Single sample gradient estimate*
  $\left( \frac{\partial L}{\partial c^v}, \frac{\partial L}{\partial c^e} \right) = \tilde{v} - \text{GM}(c_\lambda^v, c_\lambda^e)$
  **return** $\frac{1}{\lambda} \left( \frac{\partial L}{\partial c^v}, \frac{\partial L}{\partial c^e} \right)$

---

where $\tilde{v} = \text{GM}(\theta + \epsilon)$, $\lambda$ is a step size of finite difference approximation. Using a Monte Carlo approximation of the above expectation, the gradient for $\theta$ is computed as a difference of two or more pairs of perturbed graph-matching outputs. We summarize in Algorithm 1 the forward and backward steps for $c^v, c^e$.

## 4 Experiments

### 4.1 Implementation details

**Pre-training** We utilize Resnet50 [60] and Vision Transformer (ViT-B/16) [61] to train our LVM-Med. For Resnet50, we load pre-trained from ImageNet-1K [62], and SAM Encoder backbone weight [10] for ViT. The raw image is augmented to two different views by using multi-crop techniques as [14] and followed by flip (probability 50 %), color jitter, random Gaussian blur, and normalization. We trained the LVM-Med with 100 epochs on the collected dataset. The batch size of 3200 is used for ResNet50 and we reduced it to 2800 for ViT due to memory limitation. The model is optimized with Adam [63] with an initial learning rate $2 \times 10^{-3}$ and reduced halved four times. We use 16 A100-GPUs per with 80GB and complete the training process for LVM-Med with ResNet-50 in five days and LVM-Med with ViT encoder in seven days. Other competitor SSL methods as VicRegl, Twin-Barlon, Dino, etc, are initialized from ResNet-50 pre-trained ImageNet-1K and trained with 100 epochs with default settings as LVM-Med.

To balance samples among different modalities, we combine over-sampling and data augmentation to increase the total samples. Specifically, new samples from minority classes are generated by duplicating images and applying random crop operations covering $85 - 95\%$ of image regions and then rescaling them to the original resolutions. Note that these augmentations are not used in the self-supervised algorithm (operations $s, t \sim T$) to avoid generating identical distorted versions in this sampling procedure.

Table 1: Summary of datasets and downstream tasks

| Evaluation | Downstream Task Data | Modality | Nums | Task |
|---|---|---|---|---|
| Fine-Tuning | BraTS2018 [64] | 3D MRI | 285 | Tumor Segmentation |
| Fine-Tuning | MMWHS-CT [65] | 3D CT | 20 | Heart Structures Segmentation |
| Fine-Tuning | MMWHS-MRI [65] | 3D MRI | 30 | Heart Structures Segmentation |
| Fine-Tuning | ISIC-2018 [66] | 2D Dermoscopy | 2596 | Skin Leision Segmentation |
| Fine-Tuning | JSRT [67] | 2D X-ray | 247 | Multi-Organ Segmentation |
| Fine-Tuning | KvaSir [68] | 2D Endoscope | 1000 | Polyp Segmentation & Detection |
| Fine-Tuning | Drive [69] | Fundus | 40 | Vessel Segmentation |
| Fine-Tuning | BUID [70] | 2D Ultrasound | 647 | Breast Cancer Segmentation |
| Linear Evaluation & Fine-Tuning | FGADR [71] | Fundus | 1841 | DR Grading |
| Linear Evaluation & Fine-Tuning | Brain Tumor Classification | 2D MRI | 3264 | Brain Tumor Classification |
| Fine-Tuning | Multi-site Prostate MRI Segmentation [72] | 3D MRI | 116 | Prostate Segmentation |
| Fine-Tuning | VinDr [73] | 2D X-ray | 18000 | Lung Diseases Detection |

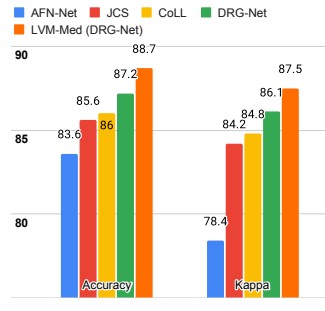

Figure 3: FGADR performance with top architectures.

**Downstream Tasks** Table 1 lists the datasets and downstream tasks used in our experiments. We cover segmentation, object detection, and image classification problems. It is important to note that in most settings, we utilize simple configurations for all datasets, skipping extra pre-processing for data augmentation. For instance, overlapping image patches with stride operations in the original samples [74] to increase training data in the Drive dataset or combining different 3D MRI modalities to fuse information [75] in the BRATS-2018 are excluded in our downstream setups.

To validate LVM-Med algorithms, we compare with 2D-SSL methods trained in our dataset and foundation models like Clip [3], Align [4], Flava [5], and SAM [6] with pre-trained ViT (Bert for

Align) taken from each method, respectively. During the downstream task, trained SSL weights are then extracted and attached in U-Net for ResNet50 backbone, TransUnet [76] for ViT, and then fine-tuned with training splits of each dataset. Depending on the downstream task's properties, we apply different image resolutions and other parameters like the number of epochs and learning rate for different data domains. Details for these configurations are presented in Appendix.

## 4.2 2D- and 3D-based segmentation

We evaluate LVM-Med on *eight* medical segmentation tasks, including five 2D-based and three 3D-based segmentation. In 2D settings, we also compare with 2D supervised architectures, such as U-Net, U-Net++, Attention U-Net, etc. These networks are initialized with ResNet-50 pre-trained ImageNet. Additionally, we investigate the prompt-based segmentation settings inspired by the current success of SAM's zero-shot learning. We utilized the ground truths and added random noise to simulate box-based user prompts as [9]. We next compare three variations of SAM: (i) freezing image and prompt encoders, only fine-tuning mask decoder; (ii) without any training and inference using box prompts; (iii) similar to (i) but replacing the original image encoder by LVM-Med's ViT architecture taken from SAM trained in our dataset.

Table 2: Performance comparison on five 2D segmentation tasks with fully fine-tuning. Results are reported with an average 2D Dice score on three trial times. The best results in each group are in bold, the overall best value, excluding prompt-based segmentation, is underlined.

| | Method | ISIC-2018 (Skin Lesion) | JSRT (Lung X-ray) | KvaSir (Polyp) | Drive (Vessel) | BUID (Breast Cancer) |
|---|---|---|---|---|---|---|
| **2D Supervised Method** | Randomly (R50) | 86.16 ± 0.14 | 93.10 ± 0.12 | 62.85 ± 1.32 | 59.82 ± 2.00 | 65.54 ± 0.21 |
| | Pre-trained ImageNet [60] | **86.87 ± 0.47** | **94.52 ± 2.66** | **83.85 ± 1.32** | 65.12 ± 1.55 | 72.64 ± 1.14 |
| | Attention-Unet [77] | 86.81 ± 0.51 | 94.47 ± 2.71 | 82.23 ± 1.41 | 65.02 ± 1.44 | 72.19 ± 1.16 |
| | U-Net ++ [78] | 86.71 ± 0.49 | 94.32 ± 2.81 | 82.23 ± 1.41 | **65.38 ± 0.78** | **73.76 ± 2.83** |
| | Trans U-Net [76] | 86.60 ± 0.82 | 89.80 ± 0.35 | 67.11 ± 0.24 | 62.63 ± 0.24 | 67.90 ± 0.40 |
| **2D-SSL on medical** | Twin-Barlon [13] | 86.01 ± 0.07 | 94.56 ± 3.09 | 83.00 ± 0.23 | 65.73 ± 1.46 | 74.46 ± 1.19 |
| | Dino [79] | 86.79 ± 0.09 | 94.84 ± 2.79 | 79.84 ± 1.62 | 65.39 ± 0.81 | 76.21 ± 0.57 |
| | SimCLR [15] | 87.28 ± 0.21 | 94.79 ± 2.93 | 82.20 ± 0.51 | 65.22 ± 2.18 | 76.52 ± 0.22 |
| | Moco-v2 [17] | 87.24 ± 0.14 | 94.05 ± 3.52 | 78.24 ± 1.35 | 64.92 ± 2.21 | 75.93 ± 1.96 |
| | Deepcluster-v2 [20] | 86.73 ± 0.42 | 94.79 ± 2.89 | 82.69 ± 0.75 | 64.14 ± 0.92 | 76.33 ± 0.99 |
| | VicRegl [14] | 86.27 ± 0.33 | 94.39 ± 3.25 | 81.93 ± 0.48 | 66.17 ± 0.27 | 75.29 ± 0.64 |
| | **LVM-Med (R50)** | **87.76 ± 0.30** | **95.13 ± 2.64** | **86.76 ± 0.94** | **66.97 ± 0.27** | **78.65 ± 0.72** |
| **Foundation Model** | Clip [3] | 85.98 ± 0.19 | 89.00 ± 1.08 | 72.63 ± 0.37 | 63.01 ± 0.36 | 70.43 ± 0.24 |
| | Flava [5] | 86.42 ± 0.10 | 90.08 ± 0.20 | 69.47 ± 0.05 | 61.09 ± 0.45 | 67.54 ± 1.17 |
| | SAM [6] | 88.17 ± 0.30 | 90.68 ± 0.40 | 70.75 ± 0.60 | 64.04 ± 0.41 | 73.07 ± 0.66 |
| | **LVM-Med (SAM's ViT)** | **88.41 ± 0.28** | **90.74 ± 0.47** | **73.10 ± 0.08** | **65.49 ± 0.12** | **77.20 ± 0.42** |
| **Prompt-based Seg.** | SAM (fixed encoder) [9] | 92.42 ± 0.12 | 92.89 ± 5.24 | 89.37 ± 0.57 | 59.74 ± 0.63 | 87.63 ± 0.67 |
| | SAM with Prompt (no-train) [6] | 55.78 ± 0.66 | 61.97 ± 4.48 | 80.77 ± 0.19 | 15.12 ± 0.24 | 78.44 ± 1.01 |
| | **LVM-Med (SAM's ViT)** | **92.48 ± 0.07** | **93.74 ± 4.06** | **90.09 ± 0.14** | **63.01 ± 0.02** | **89.69 ± 0.61** |

Table 3: 3D segmentation task performance with fine-tuning on three datasets. Results are reported with an average 3D IoU on five trial times. The best results in each group and overall are in bold and underlined.

| Method | BraTS | MMWHS-CT | MMWHS-MRI |
|---|---|---|---|
| 3D-Transformer [80] | 66.54 ± 0.40 | 67.30 ± 2.29 | 67.64 ± 2.21 |
| I3D [81] | 67.83 ± 0.75 | 76.63 ± 2.32 | 66.71 ± 1.27 |
| NiftyNet [82] | 60.78 ± 1.60 | 74.91 ± 2.78 | 64.60 ± 1.96 |
| Med3D [83] | 66.09 ± 1.35 | 75.01 ± 0.74 | 63.43 ± 0.61 |
| Model Genesis [32] | 67.96 ± 1.29 | 76.48 ± 2.89 | 74.53 ± 1.69 |
| Universal Model [84] | 72.10 ± 0.67 | 78.14 ± 0.77 | 77.52 ± 0.50 |
| TransVW [33] | 68.82 ± 0.38 | 79.74 ± 2.78 | 75.08 ± 2.04 |
| SwinViT3D [85] | 70.58 ± 1.27 | 70.19 ± 1.23 | **78.25 ± 1.66** |
| Joint-2D-3D (Deepc) [86] | **72.81 ± 0.15** | **83.58 ± 1.54** | 78.14 ± 1.32 |
| Twin-Barlon [13] | 73.30 ± 0.18 | 84.74 ± 1.01 | 76.39 ± 2.23 |
| Dino [79] | 71.72 ± 0.55 | 81.08 ± 1.62 | 70.42 ± 78.74 |
| SimCLR [15] | 73.15 ± 0.27 | 84.60 ± 1.11 | 76.54 ± 2.22 |
| Moco-v2 [17] | 71.97 ± 0.63 | 75.82 ± 4.20 | 68.29 ± 0.15 |
| Deepcluster [20] | 72.96 ± 0.51 | 84.03 ± 0.50 | **79.05 ± 1.63** |
| VicRegl [14] | 73.23 ± 0.33 | 84.72 ± 0.86 | 76.32 ± 0.78 |
| **LVM-Med (R50)** | **73.58 ± 0.14** | **84.91 ± 0.77** | 78.59 ± 0.84 |
| Clip [3] | 70.24 ± 1.23 | 78.5 ± 2.70 | 65.9 ± 3.98 |
| Flava [5] | 71.19 ± 0.48 | 78.91 ± 2.24 | 67.14 ± 1.20 |
| SAM (Encoder) [6] | 70.11 ± 1.45 | 77.8 ± 1.60 | 68.09 ± 5.49 |
| **LVM-Med (SAM's ViT)** | **71.42 ± 0.70** | **80.78 ± 1.77** | **69.36 ± 0.18** |

Table 4: In-out-distribution evaluation for the segmentation task on the Prostate dataset. Results are reported with an average 2D Dice score on three training times.

| Method | Multi-site Prostate Segmentation | | | | |
|---|---|---|---|---|---|
| | BMC (Based) | RUNMC | BIDMC | HK | Average |
| *2D Supervised* | | | | | |
| Random | 65.04 ± 2.07 | 51.44 ± 4.13 | 9.95 ± 13.56 | 12.38 ± 7.68 | 34.7 |
| Pretrained ImageNet [60] | **76.47 ± 1.26** | **62.11 ± 0.85** | **43.74 ± 4.38** | **53.90 ± 2.01** | 59.1 |
| *2D SSL on medical data* | | | | | |
| Twin-Barlon [13] | 76.28 ± 1.76 | 60.09 ± 1.98 | 32.63 ± 12.32 | 34.82 ± 15.09 | 51.0 |
| Dino [79] | 77.90 ± 1.15 | 56.90 ± 1.97 | 21.53 ± 5.54 | 30.92 ± 5.41 | 46.8 |
| SimCLR [15] | 76.51 ± 2.07 | 64.10 ± 4.53 | 32.88 ± 5.43 | 42.29 ± 5.98 | 53.9 |
| Moco-v2 [17] | 74.40 ± 0.89 | 55.49 ± 5.45 | 27.53 ± 10.18 | 13.65 ± 14.33 | 42.8 |
| Deepcluster [20] | 77.45 ± 0.35 | **64.35 ± 3.15** | 37.73 ± 8.08 | 44.95 ± 8.57 | 56.1 |
| Sway [21] | 77.59 ± 0.61 | 57.61 ± 2.16 | 38.43 ± 12.55 | 44.90 ± 4.78 | 54.6 |
| VicRegl [14] | 74.85 ± 1.13 | 54.09 ± 4.35 | 25.56 ± 5.44 | 35.45 ± 13.03 | 47.5 |
| **LVM-Med (R50)** | **80.17 ± 0.55** | 62.48 ± 2.03 | **56.76 ± 6.50** | 52.78 ± 3.04 | **63.0** |
| *Prompt-based Seg.* | | | | | |
| SAM (Fixed encoder) [9] | 95.50 ± 0.29 | 90.39 ± 0.39 | 91.41 ± 0.14 | 91.82 ± 0.26 | 92.28 |
| SAM with Prompt (no-train) [6] | 59.11 ± 1.55 | 66.95 ± 2.49 | 59.68 ± 0.49 | 57.41 ± 2.83 | 60.79 |
| **LVM-Med (SAM's ViT)** | **95.75 +- 0.06** | **90.40 +- 0.36** | **92.03 +- 0.20** | **92.75 +- 0.48** | **92.73** |

In 3D settings, we segment 2D slices and merge results for a 3D volume. We also benchmarked with 3D self-supervised methods from [86]. Tables (2) and (3) show that our two versions with ResNet-50 and Sam's ViT hold the best records in each category. For instance, we outperform 2D SSL methods trained on the same dataset, surpassing foundation models such as SAM, Flava, and Clip. In the prompt-based settings, LVM-Med also delivers better performance compared with SAM. Second, LVM-Med achieves the best overall results on *seven of eight segmentation tasks*, mostly held by

LVM-Med with ResNet-50. The improvement gaps vary on each dataset, for e.g., from $3 - 5\%$ on Kvasir and BUID compared with 2D supervised methods.

## 4.3   Linear and finetuning image classification

We analyze LVM-Med on image classification tasks using linear probing (frozen encoders) and fully fine-tuning settings, two popular evaluations used in self-supervised learning. The experiments are conducted on the FGADR Grading and Brain tumor classification tasks. Table (5) presents the average accuracy metric on three training times. LVM-Med (ResNet-50) consistently outperforms other approaches on two datasets. For example, it is better than Clip by $10.46\%$ and $8.46\%$ on FGADR and Brain Tumor datasets with linear evaluation. In the foundation model setting, LVM-Med (ViT) also improves SAM's results by $7.32\%$ and $4.69\%$ on FGADR with linear and fully-finetuning. Another point we observe is that the overall 2D-SSL methods based on ResNet-50 and trained on the collected medical dataset achieve higher accuracy than foundation models using ViT. We also compare LVM-Med with the top methods on the FGADR dataset, including AFN-Net [87], JCS [88], CoLL [89], and DRG-Net [90]. We choose the DRG-Net as the backbone and replace the employed encoder with our weights (R50). Figure (3) shows that LVM-Med hold the first rank overall.

Table 5: Performance comparison on linear evaluation and fine-tuning classification. The results are reported with average Accuracy on three training times.

| Method | Linear Evaluation (Frozen) | | Fine-tuning | |
|---|---|---|---|---|
| | FGADR | Brain Tumor Cls. | FGADR | Brain Tumor Cls. |
| Twin-Barlon [13] | $66.86 \pm 0.41$ | $63.03 \pm 0.32$ | $66.37 \pm 0.77$ | $74.20 \pm 1.38$ |
| Dino [79] | $65.98 \pm 1.91$ | $62.27 \pm 0.32$ | $67.35 \pm 1.36$ | $71.91 \pm 1.55$ |
| SimCLR [15] | $65.30 \pm 1.70$ | $62.52 \pm 1.67$ | $67.55 \pm 0.28$ | $73.52 \pm 3.56$ |
| Moco-v2 [17] | $65.98 \pm 1.04$ | $62.35 \pm 1.92$ | $67.55 \pm 1.79$ | $74.53 \pm 0.43$ |
| Deepcluster [20] | $65.34 \pm 1.93$ | $64.47 \pm 0.55$ | $67.94 \pm 1.78$ | $73.10 \pm 0.55$ |
| VicRegl [14] | $64.71 \pm 0.60$ | $59.64 \pm 1.36$ | $65.69 \pm 1.46$ | $73.18 \pm 2.03$ |
| **LVM-Med (R50)** | $\underline{68.33} \pm 0.48$ | $\underline{66.33} \pm 0.31$ | $\underline{68.32} \pm 0.48$ | $\underline{76.82} \pm 2.23$ |
| Clip [3] | $57.87 \pm 0.50$ | $57.87 \pm 0.71$ | $57.48 \pm 0.86$ | $34.86 \pm 2.27$ |
| Flava [5] | $31.87 \pm 0.69$ | $35.19 \pm 0.43$ | $57.18 \pm 0.96$ | $34.01 \pm 5.97$ |
| Algin [4] | $36.95 \pm 1.04$ | $30.71 \pm 2.35$ | $57.28 \pm 0.97$ | $63.96 \pm 0.04$ |
| SAM [6] | $55.13 \pm 0.41$ | $31.81 \pm 4.26$ | $58.75 \pm 1.32$ | $60.66 \pm 1.36$ |
| **LVM-Med (SAM's ViT)** | $62.46 \pm 0.86$ | $59.31 \pm 0.48$ | $63.44 \pm 0.73$ | $67.34 \pm 2.08$ |

Table 6: LVM-Med ablation study. Results are reported on an average of five 2D segmentation and two linear classification tasks. The two most important factors are highlighted.

| Method | Cls.(Acc) | Seg. (Dice) |
|---|---|---|
| LVM-Med (Full) | **67.47** | **83.05** |
| LVM-Med w/o second-order | 62.17 | 80.21 |
| LVM-Med w/o message passing | 65.08 | 81.19 |
| LVM-Med w/o Gumbel noise | 64.32 | 81.37 |
| LVM-Med w/o local similarity | 65.67 | 81.54 |

## 4.4   Object detection & In-out-distribution evaluation

Figure 4 indicates our performance on the object detection task using VinDr and Kvasir datasets. We use Faster R-CNN and load ResNet-50 from 2D SSL pre-trained weights. Results are presented by Average Precision with IoU=0.5 over three training times. Compared to pre-trained Imagenet, LVM-Med still outperforms by $1$-$2\%$ though overall, our improvements are smaller than image classification and segmentation tasks.

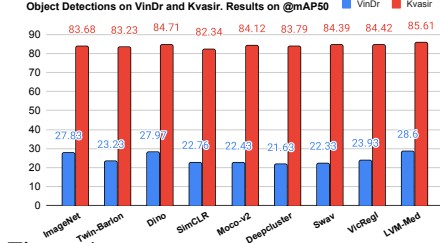

Figure 4: LVM-Med on object detection.

We also validate LVM-Med performance on the in-out-distribution setting in Table (4) using the segmentation task on the Multi-Prostate dataset. We train LVM-Med and other competitors in BMC data and use the trained models to predict the remaining datasets. Both two versions of LVM-Med with ResNet-50 and ViT, on average, surpass all baselines, which validates the potential abilities of LVM-Med for the in-out-distribution problem.

## 4.5   Ablation study

We do the following settings to evaluate the performance of components used in LVM-Med: (i) LVM-Med without using second-order graph matching conditions, i.e., only solving vertex-to-vertex correspondence problem; (ii) LVM-Med without using message passing network $g_\epsilon$ in Eq. (1) to aggregate information from local connections; (iii) LVM-Med w/o using approximate gradients from Gumbel noise in Eq. (6). For this, we add a constant value to $c^v$, $c^e$ as prior works [57, 49], and finally (iv) LMV-Med without using local similarity $c_{ia}^{lo}$ in Eq. (2). Other ablation studies are presented in Appendix. Table (6) indicates that all factors contribute to the final performance, wherein the second-order and Gumbel noise are the two most two important parts.

# 5    Conclusion

We have demonstrated that a self-supervised learning technique based on second-order graph-matching, trained on a large-scale medical imaging dataset, significantly enhances performance in various downstream medical imaging tasks compared to other supervised learning methods and foundation models trained on hundreds of millions of image-text instances. Our findings are supported by the benefits shown in two different architectures: ResNet-50 and ViT backbones, which can be used for either end-to-end or prompt-based segmentation.

**Limitations and Future Work.** We propose to investigate the following points to improve LVM-Med performance. Firstly, extending LVM-Med to a hybrid 2D-3D architecture to allow direct application for 3D medical tasks instead of 2D slices. Secondly, although LVM-Med with ViT backbone utilizes more total parameters, in some cases, it is less effective than LVM-Med ResNet-50. This raises the question of whether a novel approach could improve the performance of ViT architectures. Finally, integrating multi-modal information such as knowledge graphs, bio-text, or electronic health records for LVM-Med is also important to make the model more useful in real-world applications.

## Acknowledgements

This research has been supported by the pAItient project (BMG, 2520DAT0P2), Ophthalmo-AI project (BMBF, 16SV8639) and the Endowed Chair of Applied Artificial Intelligence, Oldenburg University. Binh T. Nguyen wants to thank the University of Science, Vietnam National University in Ho Chi Minh City for their support. Tan Ngoc Pham would like to thank the Vingroup Innovation Foundation (VINIF) for the Master's training scholarship program. The authors thank the International Max Planck Research School for Intelligent Systems (IMPRS-IS) for supporting Duy M. H. Nguyen. Mathias Niepert acknowledges funding by Deutsche Forschungsgemeinschaft (DFG, German Research Foundation) under Germany's Excellence Strategy - EXC and support by the Stuttgart Center for Simulation Science (SimTech).

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

## Supplementary Material

We present below LVM-Med pseudo-code (Section A), implementations used in downstream tasks (Section B), additional ablation studies of LVM-Med (Section C), further prompt-based segmentation results on 3D datasets, image classification benchmark (Section D), predicted masks using the user-based prompt (Section E), and finally the dataset overview (Section F).

## A  LVM-Med Pseudo-code

First, we provide a pseudo-code for training LVM-Med in Pytorch style:

```
# f_θ: encoder network, h_φ: projector network, g_ε: message passing network,
# k_nodes: number of nearest neighbors, Avg: average pooling,
# pos: position of image after transform, cos: cosine similarity,
# α: coefficient trades off between global and local costs, L₂: L2-distance,
# γ: maximum pairs are kept, select_top: select to keep the γ best matches.

for X in loader: # load a batch X = [x₁, x₂, ..., x_N] with N samples
    # apply two transformations s and t
    Xˢ, Posˢ = s(X) # Xᵏ = [x₁ᵏ, x₂ᵏ, ..., x_Nᵏ], Posᵏ = [pos₁ᵏ, pos₂ᵏ, ..., pos_Nᵏ], k ∈ {s, t}
    Xᵗ, Posᵗ = t(X)

    # compute feature representations
    Yˢ = f_θ(Xˢ); Yᵗ = f_θ(Xᵗ) # feature dimensions:NxDxRxS

    # applying projection
    Zˢ = h_φ(Avg(Yˢ)); Zᵗ = h_φ(Avg(Yᵗ)) # dimensions:NxF

    # build graph structures and message passing
    Gˢ = k-nearest-neighbor(Zˢ, k_connects)
    Gᵗ = k-nearest-neighbor(Zᵗ, k_connects)
    Ẑˢ = g_ε(Gˢ, Zˢ); Ẑᵗ = g_ε(Gᵗ, Zᵗ)

    # compute vertex and edge affinity matrices
    c_ia^v = α * cos(ẑ_i^s, ẑ_a^t) + (1 − α) * local_cost(y_i^s, y_a^t, pos_i^s, pos_a^t) # affinity x_i^s & x_a^t
    c_ia,jb^e = cos((ẑ_i^s − ẑ_j^s), (ẑ_a^t − ẑ_b^t)) # affinity between edges v_ij^s, v_ab^t
    c^v = {c_ij^v} ∈ R^{N×N}; c^e = {c_ia,jb^e} ∈ R^{|E^s||E^t|} # E^k be a set of edges in G^k, k ∈ {s, t}

    # perturbed costs with Gumbel noise
    ε, ε′ ∼ Gumbel(0, 1)
    c^v = c^v + ε; c^e = c^e + ε′

    # solving graph matching and compute loss
    v̂ = GM(c^v, c^e)
    L(v̂, v*) = v̂.(1 − v*) + v*.(1 − v̂) # compute hamming loss

    # update network
    L.backward() # approximate (∂L/∂c^v, ∂L/∂c^e) by Algorithm 1.
    Update(g_ε.params), Update(h_φ.params), Update(f_θ.params)

# define local_cost
def local_cost(y_i^s, y_a^t, pos_i^s, pos_a^t):

    # location-based local cost
    y_i,nn^s = torch.zeros_like(y_i^s)
    for r, s in R, S:
        r′, s′ = argmin((L₂(pos_i^s[r, s], pos_a^t[r′, s′]))
        y_i,nn^s[r, s] = y_a^t[r′, s′]
```

```python
y_{i_fil}^s, y_{i,nn_fil}^s = select_top (y_i^s, y_{i,nn}^s, γ)

location_cost = cos(y_{i_fil}^s, y_{i,nn_fil}^s)

# featured-based local cost
y_{i,nn}^s = torch.zeros_like(y_i^s)
for r, s in R, S:
    r', s' = argmin((L_2(y_i^s[r,s], y_a^t[r',s'])))
    y_{i,nn}^s[r,s] = y_a^t[r',s']

y_{i_fil}^s, y_{i,nn_fil}^s = select_top (y_i^s, y_{i,nn}^s, γ)

feature_cost = cos(y_{i_fil}^s, y_{i,nn_fil}^s)

return 0.5*(location_cost + feature_cost)
```

We trained LVM-Med with graph size of 16 nodes, each node connected to the top 5 nearest neighbors after using kNN, $\lambda$ value in Algorithm 1 is $80$, and $\alpha = 0.8$ for associating global- and local-based similarities when computing $c_{ij}^v$. The size of projector $h_\phi$ is $2048 \times 128$ for ResNet-50, and $768 \times 128$ for ViT. We configure the message passing network $g_\theta$ with two convolutional layers of size 128. For the user-based prompt version, because the SAM model [9] requires an input of shape $256 \times 14 \times 14$ for the mask decoder part, we add two additional convolutional layers with a kernel size of 1 and 3 at the end of ViT backbone to convert from shape $768 \times 14 \times 14$ to the target shape.

## B  Downstream task setups

### B.1  Downstream tasks

**Segmentation tasks**  On 2D-based segmentation tasks, we employ U-Net architecture [91] and load ResNet-50 [60] trained by self-supervised learning algorithms as network backbones. With foundation models, we use TransUnet [76] and take pre-trained ViT models as the backbones. For the prompt-based segmentation, we follow the architecture of SAM [6] consisting of encoder, prompt, and mask decoder layers. We also fine-tune SAM where encoder and prompt networks are frozen, only learning decoder layers [9]. Our LVM-Med for prompt-based setting is similar to [9] except that we substitute SAM's encoders with our weights. We utilize Adam optimizer for all experiments and train architectures with Dice and Cross-Entropy loss [92]. We also normalize the norm-2 of gradient values to stabilize the training step to maximize 1. Table 7 summarizes each dataset's learning rate, number of epochs, and image resolution.

On 3D-based segmentations, we reformulate these tasks as 2D segmentation problems and make predictions on 2D slices taken from 3D volumes. Furthermore, we apply balance sampling to select equally 2D slices covering target regions and other 2D slices, not including the ground truth. Table 8 presents configurations used for 3D datasets; other settings are identical to 2D cases.

Table 7: Configurations for training 2D segmentation tasks

| | ISIC-2018 (Skin Lesion) | JSRT (Lung X-ray) | KvaSir (Polyp) | Drive (Vessel) | BUID (Breast Cancer) |
|---|---|---|---|---|---|
| **ResNet-50.** | lr = $10^{-4}$, epochs 35
shape $512 \times 512$
batch size 16 | lr = $10^{-3}$, epochs 50
shape $224 \times 224$
batch size 32 | lr = $10^{-3}$, epochs 35
shape $224 \times 224$
batch size 64 | lr = $10^{-3}$, epochs 50
shape $224 \times 224$
batch size 16 | lr = $10^{-4}$, epochs 50
shape $256 \times 256$
batch size 8 |
| **Foundation Model** | lr = $10^{-4}$, epochs 100
shape $512 \times 512$
batch size 16 | lr = $10^{-3}$, epochs 200
shape $224 \times 224$
batch size 32 | lr = $10^{-3}$, epochs 200
shape $224 \times 224$
batch size 64 | lr = $10^{-3}$, epochs 200
shape $224 \times 224$
batch size 16 | lr = $10^{-4}$, epochs 200
shape $256 \times 256$
batch size 8 |
| **Prompt-based Seg.** | lr = $10^{-4}$, epochs 50
shape $1024 \times 1024$
batch size 16 | lr = $3 \times 10^{-4}$, epochs 50
shape $1024 \times 1024$
batch size 16 | lr = $3 \times 10^{-4}$, epochs 20
shape $1024 \times 1024$
batch size 16 | lr = $3 \times 10^{-4}$, epochs 100
shape $1024 \times 1024$
batch size 16 | lr = $10^{-4}$, epochs 20
shape $1024 \times 1024$
batch size 16 |

Table 8: Configurations for 3D-based-segmentation tasks

|  | BraTS | MMWHS-CT | MMWHS-MRI | BMC |
|---|---|---|---|---|
| **ResNet50** | lr = $15 \times 10^{-4}$, epochs 20
shape $224 \times 224$
batch size 128 | lr = $10^{-3}$, epochs 20
shape $224 \times 224$
batch size 64 | lr = $15 \times 10^{-4}$, epochs 30
shape $224 \times 224$
batch size 64 | lr = $10^{-3}$, epochs 30
shape $224 \times 224$
batch size 64 |
| **Foundation Model** | lr = $10^{-4}$, epochs 100
shape $224 \times 224$
batch size 16 | lr = $10^{-4}$, epochs 100
shape $224 \times 224$
batch size 16 | lr = $10^{-4}$, epochs 100
shape $224 \times 224$
batch size 16 | lr = $10^{-4}$, epochs 100
shape $224 \times 224$
batch size 16 |
| **Prompt-based Seg.** | lr = $3 \times 10^{-5}$, epochs 30
shape $1024 \times 1024$
batch size 16 | lr = $5 \times 10^{-5}$, epochs 30
shape $1024 \times 1024$
batch size 16 | lr = $3 \times 10^{-5}$, epochs 30
shape $1024 \times 1024$
batch size 16 | lr = $3 \times 10^{-4}$, epochs 50
shape $1024 \times 1024$
batch size 16 |

**Image classification tasks**  We take the feature embedding outputs of each architecture and build one fully connected layer to produce desired classes for image classification tasks. We freeze the encoder layers for the linear evaluation and only train the fully connected layer. For the fully-finetuning, the whole network is trained. The Adam optimizer [63] with cross-entropy loss function and learning rates $\{5 \times 10^{-4}, 10^{-3}\}$ are used for Brain Tumor and FGADR, respectively. To benchmark LVM-Med with other state-of-the-art methods on FGADR (Figure 3 in paper), we follow the settings of DRG-Net [90] and change their encoder layers by our networks.

**Object detection**  We use Faster-RCNN [93] for object detection tasks. The ResNet-50 of Faster-RCNN is replaced by pre-trained weights. In the Vin-Dr dataset, there is a total of $14$ objects for, e.g., Aortic enlargement, Atelectasis, Calcification, etc. We use image resolutions of $512 \times 512$, Adam solver, and learning rate $10^{-4}$ in 40 epochs. In the Kvasir dataset for polyp detection, we also resize images to a fixed size of $512 \times 512$, employ the Adam optimizer with learning rate $2.5 \times 10^{-4}$ and batch size $8$.

# C   LVM-Med ablation studies

## C.1   Graph sizes and $\lambda$ in backpropagation

We provide in Figure 5 and Figure 6 LVM-Med performance when changing the number of nodes in graph construction steps $G^s, G^t$ and $\lambda = 80$ used in Algorithm 1 in the backpropagation step. The results are reported on the average Dice score of five 2D segmentation tasks and the average accuracy of two linear classifications on FGADR and Brain Tumor Classification. Figure 5 indicates that 16 is the best value for both classification and segmentation. Increasing the graph's nodes tends to decrease classification performance.

Figure 6 compared different values for $\lambda \in \{70, 80, 90, 100\}$. We observe that $\lambda = \{80, 90\}$ achieve good results for linear classification tasks though $\lambda = \{90, 100\}$ decreases segmentation performance.

## C.2   Performance on large- and small-scale

We investigate LVM-Med performance when reducing the number of datasets in the pre-training step. Especially, we trained LVM-Med on a *small-scale* with four datasets: LUNA2016 [94], LiTS2017 [95], BraTS2018 [64], and MSD (Heart) [96]. We compare this version with our default settings trained on $55$ datasets (Section F). Two models are evaluated on dice scores of five 2D segmentation tasks, the accuracy metric of two linear image classifications, and mAP50 of two object detection tasks on VinDr and Kvasir detection. Table 9 shows that LMV-Med full leads to better performance overall, especially with the classification settings; the improvement gap is around $3.6\%$. In summary, we conclude that LVM-Med is beneficial when training in large-scale medical settings.

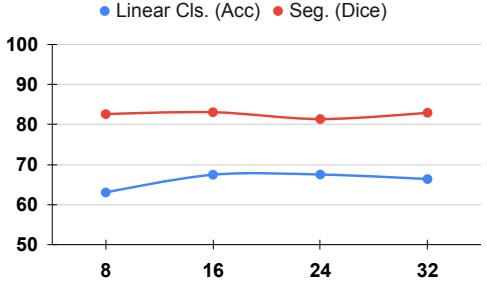

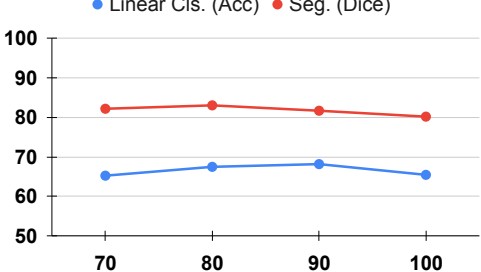

Figure 5: LVM-Med performance when varying the number of nodes in graph construction.

Figure 6: LVM-Med performance when varying the $\lambda$ in backpropagation step.

## C.3  Performance on weighting global and local similarities

We test with different $\alpha = \{0.7, 0.8, 0.9\}$ which used to fuse global- and local-based similarities $c_{ij}^{v}$. Table 9 demonstrates that $\alpha = 0.8$ is generally the best value in average across segmentation, classification, and object detection tasks.

Table 9: LVM-Med ablation studies trained with full data, small-scale, and different hyper-parameter $\alpha$ fusing global- and local-based similarities. Results are reported on an average of five 2D segmentation, two linear classifications, and two object detection tasks. The most impacted factors are highlighted.

| Method | Cls.(Acc) | Seg. (Dice) | Detect. (mAP50) |
|---|---|---|---|
| LVM-Med (full, $\alpha = 0.8$) | **67.47** | **83.05** | 57.1 |
| LVM-Med (small-scale, $\alpha = 0.8$) | 63.83 | 81.97 | 56.03 |
| LVM-Med (full, $\alpha = 0.7$) | 65.89 | 82.20 | 56.49 |
| LVM-Med (full, $\alpha = 0.9$) | 65.03 | 81.09 | **57.14** |

## C.4  Computational complexity

We present a parameter comparison of LVM-Med with other foundation models in Table 10. Our LVM-Med model, based on ResNet-50, has significantly fewer parameters, approximately 3-4 times smaller than models such as Flava or SAM, while still maintaining competitive performance. When utilizing the ViT encoder pre-trained by the SAM method, LVM-Med's parameters are comparable to the Flava model and slightly higher than Clip and Align by 1.03 and 1.43 times, respectively. However, it is important to note that both LVM-Med and SAM outperform these models by a significant margin in various settings.

Table 10: Computational complexity of our approaches and other foundation models.

| Method | LVM-Med (R50) | LVM-Med (ViT) | Clip [3] | Flava [5] | Align [4] | SAM (Encoder) [6] |
|---|---|---|---|---|---|---|
| **#Param** | 25.55 M | 88.88 M | 85.80 M | 86.39 M | 62.14 M | 88.88 M |

## D  Prompt-based segmentation on 3D datasets and classification tasks

We provide additional results for LVM-Med on 3D-based prompt segmentation and image classification tasks with several fully connected layers.

Table 12: Comparing SSL approaches and Foundation models on classification tasks with two evaluation protocols, Linear evaluation and full Fine-tuning. Settings used with several fully connected layers are in cyan. The best results in 2D-SSL and foundation models (two fully connected layers) are in bold; the best results overall are in bold and underlined.

| | Method | Linear Evaluation (Frozen) | | Fine-tuning | |
|---|---|---|---|---|---|
| | | FGADR (DR Grading) | Brain Tumor Class. | FGADR (DR Grading) | Brain Tumor Class. |
| **2D-SSL on medical** | Twin-Barlon [13] | $66.86 \pm 0.41$ | $63.03 \pm 0.32$ | $66.37 \pm 0.77$ | $74.20 \pm 1.38$ |
| | Dino [79] | $65.98 \pm 1.91$ | $62.27 \pm 0.32$ | $67.35 \pm 1.36$ | $71.91 \pm 1.55$ |
| | SimCLR [15] | $65.30 \pm 1.70$ | $62.52 \pm 1.67$ | $67.55 \pm 0.28$ | $73.52 \pm 3.56$ |
| | Moco-v2 [17] | $65.98 \pm 1.04$ | $62.35 \pm 1.92$ | $67.55 \pm 1.79$ | $74.53 \pm 0.43$ |
| | Deepcluster [20] | $65.34 \pm 1.93$ | $64.47 \pm 0.55$ | $67.94 \pm 1.78$ | $73.10 \pm 0.55$ |
| | VicRegl [14] | $64.71 \pm 0.60$ | $59.64 \pm 1.36$ | $65.69 \pm 1.46$ | $73.18 \pm 2.03$ |
| | **LVM-Med (R50)** | $\underline{\mathbf{68.33}} \pm \mathbf{0.48}$ | $66.33 \pm 0.31$ | $68.32 \pm 0.48$ | $76.82 \pm 2.23$ |
| | | $66.67 \pm 0.84$ | $\underline{\mathbf{74.70}} \pm \mathbf{0.84}$ | $\underline{\mathbf{70.58}} \pm \mathbf{0.36}$ | $\underline{\mathbf{78.77}} \pm \mathbf{0.78}$ |
| **Foundation Model** | Clip [3] | $57.87 \pm 0.50$ | $57.87 \pm 0.71$ | $57.48 \pm 0.86$ | $34.86 \pm 2.27$ |
| | | $62.66 \pm 0.36$ | $\mathbf{67.85} \pm \mathbf{0.23}$ | $56.21 \pm 1.86$ | $21.74 \pm 1.14$ |
| | Flava [5] | $31.87 \pm 0.69$ | $35.19 \pm 0.43$ | $57.18 \pm 0.96$ | $34.01 \pm 5.97$ |
| | | $32.84 \pm 0.12$ | $24.45 \pm 4.30$ | $56.01 \pm 0.86$ | $33.67 \pm 8.11$ |
| | Algin [4] | $36.95 \pm 1.04$ | $30.71 \pm 2.35$ | $57.28 \pm 0.97$ | $63.96 \pm 0.04$ |
| | | $38.12 \pm 1.45$ | $30.34 \pm 1.35$ | $57.87 \pm 0.90$ | $61.42 \pm 0.25$ |
| | SAM [6] | $55.13 \pm 0.41$ | $31.81 \pm 4.26$ | $58.75 \pm 1.32$ | $60.66 \pm 1.36$ |
| | | $57.48 \pm 0.24$ | $36.89 \pm 1.61$ | $58.75 \pm 0.99$ | $60.07 \pm 0.31$ |
| | **LVM-Med (SAM's ViT)** | $62.46 \pm 0.86$ | $59.31 \pm 0.48$ | $63.44 \pm 0.73$ | $67.34 \pm 2.08$ |
| | | $\mathbf{63.83} \pm \mathbf{1.36}$ | $64.13 \pm 1.14$ | $\mathbf{59.04} \pm \mathbf{0.14}$ | $\mathbf{64.97} \pm \mathbf{2.71}$ |

## D.1 Promt-based Segmentation on 3D datasets

We perform experiments on three 3D datasets in Table 11, including BraTS, MMWHS-MRI, and MMWHS-CT. The setup for box prompts follows 2D segmentation cases. We discover that the LMV-Med in 3D cases consistently improves the performance of fine-tuned SAM [9] as in 2D settings and attains a large margin compared with SAM without training [6]. This evidence thus confirms that LVM-Med is also effective under prompt-based scenarios.

Table 11: Prompt-based segmentation on 3D datasets.

| | Method | BraTS | MMWHS-MRI | MMWHS-CT |
|---|---|---|---|---|
| **Prompt-based Seg.** | SAM (fixed encoder) [9] | $85.37 \pm 0.07$ | $77.64 \pm 1.14$ | $76.61 \pm 1.91$ |
| | SAM with Prompt (no-train) [6] | $38.97 \pm 0.21$ | $59.74 \pm 0.76$ | $50.25 \pm 0.33$ |
| | **LVM-Med (SAM's ViT)** | $\mathbf{85.76} \pm \mathbf{0.07}$ | $\mathbf{78.91} \pm \mathbf{0.80}$ | $\mathbf{78.03} \pm \mathbf{0.93}$ |

## D.2 Image classification

We aim to inspect whether foundation models improve their performance given more fully connected layers for image classification tasks with both frozen encoders or fully fine-tuning. For each method in this category and our LVM-Med (ResNet-50 and ViT), we configure two fully connected layers with sizes $512 - 256$ and $512 - 128$ for the Brain and FGADR respectively that map from the output dimension of each network to a number of desired classes. Table 12 presents obtained results where new settings are highlighted in color. We notice the following points. (i) Firstly, using more fully connected layers tends to improve the performance of foundation models, especially on linear evaluation. For e.g., the Clip increases from $4.79\% - 9.98\%$ on FGADR and Brain Tumor classification tasks, respectively. Similarly, our LVM-Med with SAM's ViT also achieves better results by approximately $1.37\%$ and $4.82\%$ on those tasks. (ii) Secondly, LVM-Med overall attains the best results in four settings using linear or several fully connected layers with ResNet-50. LVM-Med with ViT architecture also delivers the best records on three of four test cases compared with foundation models.

## E Visualizing results

We provide qualitative results for prompt-based segmentation in Figure 7. We compare three approaches, including (i) the standard SAM without fine-tuning [6] (second column), (ii) SAM

with encoders and prompt networks are frozen, and only decoder layers are trained as [7] (third column), and (iii) a similar setting as (ii) but encoders taken from LVM-Med version with SAM's ViT architecture (fourth column). For all methods, we simulate box-based prompts using the ground-truth masks and define boxes covering those target regions perturbed by offset values.

Figure 7 demonstrates that the original SAM is prone to generate useless predictions (top and bottom rows) or less precise boundaries. In contrast, updated SAM and LVM-Med produce more accurate results, confirming the importance of fine-tuning to achieve adequate results. Figures in the third and fourth columns also illustrate that SAM tends to over-segment or lacks structures on an object's edges in several cases, while LVM-Med is more stable in those situations (red arrows).

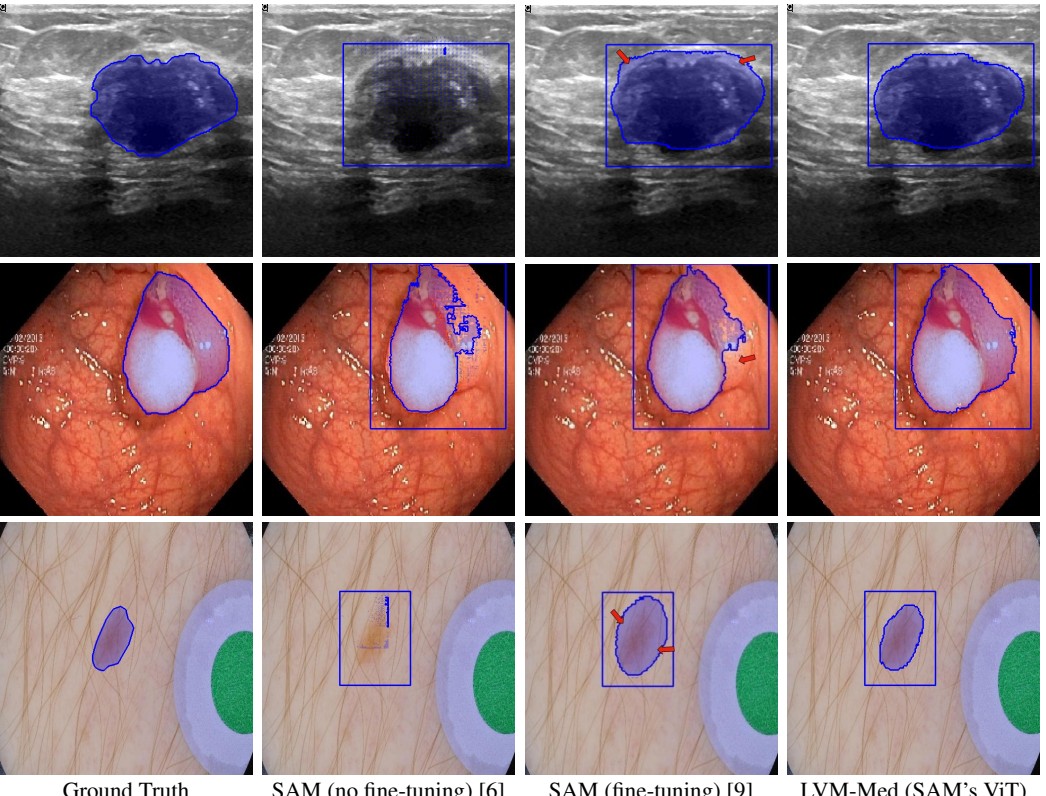

|Ground Truth|SAM (no fine-tuning) [6]|SAM (fine-tuning) [9]|LVM-Med (SAM's ViT)|

Figure 7: Visualizing prompt-based predictions on three datasets: BUID, Kvasir, and ISIC. Red arrows show differences between SAM (fine-tuning) and LVM-Med using SAM's ViT architecture. Best viewed in color with **zoom**.

## F  Dataset overviews

Table 13 overviews the dataset used in our study. For each dataset, we provide its modality, data dimension, and the total of samples. If the training/testing rate is available (column **Train/Test Rate**), we utilize all training data; otherwise, we sample $20\%$ total samples to avoid potential test data leaking for downstream tasks used in the pre-training step. For datasets whose data dimensions are 3D volumes, we sample 2D slices from those formats. Some datasets, such as MSD or ADNI, comprise different sub-datasets inside; we consider these sub-sets as independent ones to avoid confusion during the training steps. In summary, a total of 55 datasets are used with approximately $40\%$ in 3D datasets and $60\%$ in 2D images as presented in Figure 8. Moreover, we also outline ratios between distinct data modalities such as MRI, CT, X-

ray, grayscale types such as Ultrasound, OCT, and finally, color images depicted in Figure 9.

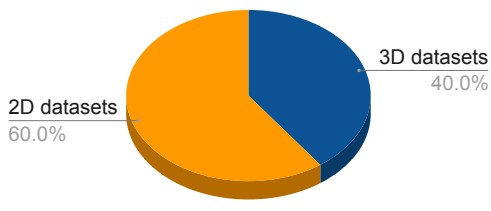

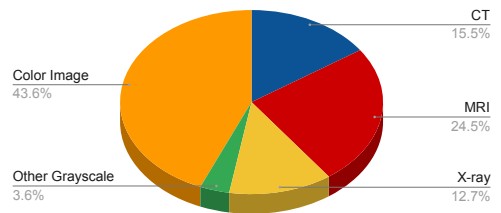

Figure 8: Pie chart illustrating the ratio of the number of datasets with the 3D and 2D dimension.

Figure 9: Pie chart illustrating the ratio of different data modalities in our collected dataset.

Table 13: Overview of our collected medical dataset

| No | Data Name | Topic | Disease | Modality | Dimension | Train/Test Rate? | Total |
|----|-----------|-------|---------|----------|-----------|------------------|-------|
| 1 | HyperKvasir [97] | Polyp | Pathological classification | Color images | 2D | Yes | 110079 |
| 2 | PatchCamelyon [98, 99] | Cells | Histopathologic scans of lymph node sections. | Color images | 2D | Yes | 327680 |
| 3 | BraTS2018 [100, 101, 64] | Brain | Tumor Segmentation | MRI | 3D | No | 760 |
| 4 | HNSCC [102] | Head Neck | No Label | CT | 3D | No | 155 |
| 5 | LiTS2017 [103] | Liver | Segmentation of Liver and Tumor Lesions | CT | 3D | No | 200 |
| 6 | MSD-Heart [96] | Heart | Heart Segmentation | MRI | 3D | No | 30 |
| 7 | MSD-Liver [96] | Liver | Liver Segmentation | MRI | 3D | No | 201 |
| 8 | MSD-Lung [96] | Lung | Lung Segmentation | MRI | 3D | No | 96 |
| 9 | MSD-Pancreas [96] | Pancreas | Pancrea Segmentation | MRI | 3D | No | 420 |
| 10 | MSD-HepaticVessel [96] | Hepatic Vessel | Hepatic Vessel Segmentation | MRI | 3D | No | 443 |
| 11 | MSD-Spleen [96] | Spleen | Spleen Segmentation | MRI | 3D | No | 61 |
| 12 | MSD-Colon [96] | Colon | Colon Segmentation | MRI | 3D | No | 190 |
| 13 | OPC-Radiomics [104–106] | Oropharynx | No Label | CT | 3D | No | 120 |
| 14 | Osteosarcoma-UT [107, 108, 105] | Osteosarcoma | No Label | Color images | 2D | No | 547 |
| 15 | Pancreas-CT [109, 110, 105] | Pancreas | No Label | CT | 3D | No | 16 |

| No | Data Name | Topic | Disease | Modality | Format | Default Train/Test Rate | Total |
|---|---|---|---|---|---|---|---|
| 16 | Pelvic-Reference-Data [108, 105] | Pelvic | No Label | CT | 3D | No | 12 |
| 17 | ProstateX [111, 112, 105] | Prostate | The clinical significance of prostate lesions prediction | MRI | 3D | No | 40 |
| 18 | TCGA-CESC [113, 105] | Cervical | No Label | Color images | 2D | No | 3977 |
| 19 | TCGA-COAD [114, 105] | Colon | No Label | Color images | 2D | No | 1644 |
| 20 | TCGA-ESCA [115, 105] | Cuticle | No Label | Color images | 2D | No | 4427 |
| 21 | TCGA-KICH [116, 105] | Kidney | No Label | Color images | 2D | No | 2192 |
| 22 | TCGA-KIRC [117, 105] | Kidney | No Label | Color images | 2D | No | 34108 |
| 23 | TCGA-READ [114, 105] | Rectum | No Label | Color images | 2D | No | 248 |
| 24 | TCGA-SARC [118, 105] | Sarcoma | No Label | Color images | 2D | No | 624 |
| 25 | TCGA-THCA [114, 105] | Thyroid | No Label | Color images | 2D | No | 665 |
| 26 | VinDr [73] | Lung | Abnormal Disease Classification | X-ray | 2D | No | 18000 |
| 27 | LUNA2016 [94] | Lung | Nodule Detection and False Positive Reduction | CT | 3D | No | 49386 |
| 28 | BCCD [119] | Cells | Blood cell detection | Color images | 2D | No | 364 |
| 29 | C-NMC_Leukemia [120, 121] | Cells | Leukemia detection | Color images | 2D | Yes | 12529 |
| 30 | CBIS-DDSM [122, 123] | Breast | Breast Cancer Classification | X-ray | 2D | No | 6774 |
| 31 | COVIDx [124] | Lung | Covid-19 Detection | X-ray | 2D | Yes | 194922 |
| 32 | Heidelberg OCT [125] | Eye | OCT Imaging Classification | OCT | 2D | Yes | 84495 |
| 33 | m2caiSeg [126] | Laparoscopic | Semantic Segmentation Laparoscopic | Color images | 2D | Yes | 614 |
| 34 | NuCLS [127] | Nucleus | Nucleus Segmentation Detection / Classification | Color images | 2D | Yes | 1744 |
| 35 | SARAS-MESAD [128][129][130] | Prostatectomy Procedures | Action classification in Prostatectomy Surgey | Color images | 2D | Yes | 29454 |
| 36 | Shoulder X-ray images from Sun Yat-sen Memorial Hospital [131] | Shoulder | Shoulder X-ray Classification | X-ray | 2D | Yes | 1049 |

| No | Data Name | Topic | Disease | Modality | Format | Default Train/Test Rate | Total |
|----|-----------|-------|---------|----------|--------|-------------------------|-------|
| 37 | Shenzhen Hospital X-ray Set [132] | Lung | Lung segmentation | X-ray | 2D | No | 566 |
| 38 | ADNI 1.5T [133, 134] | Brain | Alzheimer's Disease Classification | MRI | 3D | No | 639 |
| 39 | ADNI 3T [133, 134] | Brain | Alzheimer's Disease Classification | MRI | 3D | No | 119 |
| 40 | AML-Cytomorphology [135, 136, 105] | Cell | Peripheral blood smears | Color images | 2D | No | 18365 |
| 41 | APTOS 2019 [137] | Eye | Severity of diabetic retinopathy Classification | Color images | 2D | No | 3662 |
| 42 | BCSS [138] | Cells | Breast cancer semantic segmentation | Color images | 2D | No | 151 |
| 43 | Dental Panoramic [139] | Tooth | Mandible segmentation | X-ray | 2D | No | 116 |
| 44 | HC18 [140] | Fetal | Fetal head circumference (HC) | Ultrasound | 2D | No | 999 |
| 45 | Hippseg 2011 [141] | Brain | Hippocampus Segmentation | MRI | 3D | No | 3050 |
| 46 | ISIC Challenge 2019 [142] | Skin | Skin Cancer Classification | Color images | 2D | No | 25331 |
| 47 | KiTS19-21 [143] | Kidney | Kidney Segmentation | CT | 3D | No | 45424 |
| 48 | Kvasir v2 [144] | Gastrointestinal | Gastrointestinal cancer image classification | Color images | 2D | No | 6000 |
| 49 | LHNCBC Malaria [145] | Cells | Malaria Classification | Color images | 2D | No | 27560 |
| 50 | MitoEM [146] | Cells | Mitochondria Instance Segmentation | MRI/CT | 3D | No | 1000 |
| 51 | MLL Bone Marrow [147] | Cells | Blood cell classification | Color images | 2D | No | 171374 |
| 52 | MMWHS-CT[65] | Heart | Sub-structure Heart segmentation | CT | 3D | Yes | 40 |
| 53 | MMWHS-MRI [65] | Heart | Sub-structure Heart segmentation | MRI | 3D | Yes | 40 |
| 54 | RSNA Bone Age [148] | Bone | Bone age prediction | X-ray | 2D | No | 12611 |
| 55 | EyePACS [149] | Eye | Diabetic Retinopathy Detection | Color Images | 2D | Yes | 88702 |

