# OpenReview forum: "LVM-Med: Learning Large-Scale Self-Supervised Vision Models for Medical Imaging via Second-order Graph Matching"
_NeurIPS.cc/2023/Conference — NeurIPS 2023 poster_

### Official Review · Reviewer_yfed · 2023-07-02

**Soundness:** 3 good
**Presentation:** 3 good
**Contribution:** 3 good
**Rating:** 7
**Confidence:** 4

**Summary:**

This research paper focuses on LVM-Med, a self-supervised learning (SSL) technique designed for medical imaging tasks. Based on a second-order graph matching strategy, LVM-Med is trained on a large-scale medical imaging dataset. The researchers found that the method significantly improves performance on a variety of downstream medical imaging tasks compared to other supervised learning methods and foundation models trained on large quantities of image-text data. These findings were consistent across two different architectures: ResNet-50 and Vision Transformer (ViT).

**Strengths:**

- The LVM-Med method was evaluated on various tasks, including segmentation, object detection, and image classification. The results were compared to foundational models like Clip, Align, Flava, and SAM. It performed particularly well on eight medical segmentation tasks, outperforming both 2D SSL methods trained on the same dataset and foundational models.
- The training of LVM-Med on a large-scale medical imaging dataset indicates its capacity to handle and learn from large amounts of data, which is often a requirement in the field of medical imaging.
- The LVM-Med model outperforms both supervised learning methods and foundation models trained on hundreds of millions of image-text instances. This includes a variety of popular models such as Clip, Align, Flava, and SAM.
- The results show that LVM-Med performs well in in-out-distribution settings, implying a certain level of robustness and generalizability.
- LVM-Med provides benefits in both end-to-end and prompt-based segmentation tasks. This flexibility can be particularly useful in real-world applications, where various segmentation scenarios may be encountered.
- The authors also conducted an ablation study, experimenting with variations in LVM-Med's configuration to assess the importance of various components in the overall performance. It was concluded that all factors contribute to the final performance, with the second-order graph matching and Gumbel noise being the most significant.

**Weaknesses:**

- Only 2D backbone is conducted: The LVM-Med model focuses primarily on 2D backbone. Author should provide more discussion about the challenge for applying this framework in 3D backbone.
- The LVM-Med model is trained on a large-scale dataset, how can author endure the testing dataset is not leaky in training set. Moreover, the use of a single dataset could potentially limit its generalizability. Ensuring robust performance across diverse datasets from different sources is critical.
- The model's performance benefits seem to be tied to its training on a large-scale medical imaging dataset. If such a dataset is not available, the performance of the model might be significantly diminished.

**Questions:**

Please address the comment in weakness part.
In addition, though the ViT architecture has more total parameters, in some cases, it is less effective than LVM-Med ResNet-50. More research and discussion would be needed to fully understand this performance discrepancy.

**Limitations:**

No limitation on limitations and broader societal impacts.

---

> ### Author Rebuttal · Authors · 2023-08-08
>
> Thank you very much for your positive and constructive feedback!
>
>
> **Question 1: Add more discussion about the challenge of applying this framework in 3D backbone to the paper.**
>
> Thank you for the suggestions. Due to the large workload in dataset collection and experiments, we restricted this work to solving 2D backbone-related downstream tasks. For the 3D backbone for video or 3D volume classification, it requires a dedicated study to investigate further how to extend our proposed algorithm and design optimal architectures. We will add more discussion toward directions in the Limitations and Future Work Section, which includes the following points:
>
> - Extend the architecture in LVM-Med to dynamically receive temporal inputs like frames in videos or consecutive slices of 3D volumes rather than treating them independently (1).
>
> - (1) leads to how to modify the LVM-Med’s graph-matching to encompass combinatorial constraints among internal sections inside 3D  volumes and between slices across different inputs. One possible solution is to leverage the deformable attention mechanisms, which only focus on a flexible small set of major slices conditioned on input data [1], resulting in saving computational complexity and permitting handling the multi-scale feature maps.
>
> [1] Zhuofan Xia et al., “Vision Transformer with Deformable Attention,” CVPR 2022.
>
> &nbsp;
> &nbsp;
> &nbsp;
>
> **Question 2: How can avoid the testing dataset not leaky in the training set?**
>
> We thank the reviewer for the great questions. In particular, to use LVM-Med and avoid potential testing data leaking during the training steps, there are some typical cases:
> - If the dataset used in downstream tasks does not belong to our 55 collected ones, users can freely download and apply our models.
> - If the dataset belongs to the collection, the user needs to check whether it has default training or testing. If some splitting is available, we follow them and only use the training samples; otherwise, the user needs to exclude the index of images (we sampled 20% of total images for training) trained by LVM-Med. We will release exact information for the later cases in our code repository.
>
> &nbsp;
> &nbsp;
> &nbsp;
>
> **Question 3: The use of a single dataset could potentially limit its generalization. Ensuring robust performance across diverse datasets from different sources is critical**
>
> In our large-scale medical datasets, collected images cover diverse body organs and data molarities, as demonstrated in Figure 1 (main paper) and Table 7 (Appendix). It is not from a single large dataset. While some datasets have a large number of samples, we address it by applying some balancing strategies in each mini-batch  during the training procedure to avoid a potential bias toward a specific domain.
>
> In downstream experiments, we selected various types of data modalities, encompassing MRI, CT scans, X-rays, ultrasound images, and color images. Our settings also span eleven distinct organ structures (including tumors, heart, skin, brain, etc). The majority of these experiments have revealed favorable outcomes with LVM-Med compared to alternative reference models. These consistent records, therefore, highlight the strength and adaptability of LVM-Med across different areas of expertise.
>
> &nbsp;
> &nbsp;
> &nbsp;
>
> **Question 4: The model's performance benefits seem to be tied to its training on a large-scale medical imaging dataset. If such a dataset is not available, the performance of the model might be significantly diminished**
>
> We agree with the reviewer that the achieved performance has strongly aligned with the scale of the dataset we collected. This aspect indeed stands out as our primary discovery and is one of the most critical LVM-Med contributions alongside the novel self-supervised learning approach based on graph matching. We believe that our research will catalyze future investigations into the creation of expansive medical datasets and a deeper exploration of practical applications in real-world medical scenarios, thereby pushing the boundaries of utilizing machine learning within medicine.
>
>
> &nbsp;
> &nbsp;
> &nbsp;
>
> **Question 5: ViT architecture, in some cases, is less effective than LVM-Med ResNet-50. More research and discussion would be needed to understand this performance discrepancy fully.**
>
> In the **Limitations and Future Work Section**, we discussed this concern, i.e., ViT architecture performance with end-to-end learning. To gain a more comprehensive understanding of this phenomenon and effectively tackle it, we propose the need for additional experimentation. For instance, optimizing hyperparameters like projector heads and token feature dimensions through grid searches during pre-training is important. Another potential solution is integrating the trained architecture and fine-tuning it efficiently for downstream tasks. While our study explores the latter approach, we intend to delve into this aspect and update our findings in publicly available code repositories. For the pre-training hypothesis, we suggest it as a future work for investigation, given the huge amounts of required computations.

---

> > ### Comment · Reviewer_yfed · 2023-08-13
> >
> > Rebuttal solves my question well. In the context of training large-scale 3D medical data, there are some prior works. To enhance reader comprehension and provide a comprehensive outlook, it would be nice to add these into Limitations and Future Work Section.
> >
> > [1] Liu, Jie, et al. "CLIP-Driven Universal Model for Organ Segmentation and Tumor Detection." arXiv preprint arXiv:2301.00785 (2023).
> > [2] Ulrich, Constantin, et al. "MultiTalent: A Multi-Dataset Approach to Medical Image Segmentation." arXiv preprint arXiv:2303.14444 (2023).
> > [3] Wasserthal, et al. "TotalSegmentator: robust segmentation of 104 anatomical structures in CT images." arXiv preprint arXiv:2208.05868. (2023).

---

> > > ### Author Response · Authors · 2023-08-14
> > > **Thank you**
> > >
> > > Dear Reviewer,
> > >
> > > Thank you very much for reading our response and giving additional feedback.
> > >
> > > We will add the references suggested by the reviewer in the revision.

---

### Official Review · Reviewer_xKHb · 2023-07-03

**Soundness:** 3 good
**Presentation:** 3 good
**Contribution:** 4 excellent
**Rating:** 8
**Confidence:** 4

**Summary:**

This paper proposes a self-supervised pre-training strategy for medical imaging using a graph matching approach. Each unlabeled image is transformed via a pair of data augmentations and then processed via an encoder network. The augmented pair of images become vertices in a pair of graphs, with vertex features being the encoder outputs and edge connections selected via k-nn. A graph convolutional network is trained for vertex-to-vertex matching of the extracted pair of graphs. The training objective incorporates global and local similarity learning over spatial features along with a second-order edge similarity cost. Due to the combinatorial nature of the objective, gradients for backpropagation are approximated via Implicit MLE. Pre-training is performed over a large scale dataset comprising 55 publicly available datasets and used for multiple downstream fine-tuning tasks including segmentation, detection and classification.

**Strengths:**

The proposed graph matching technique for self-supervised learning is a novel and significant contribution. Abundant experiments demonstrate the generalizability over downstream tasks, with error bars also included.

**Weaknesses:**

Some related works on graph matching in computer vision are missing in Section 2.3.

Doi et al. Detecting Object-Level Scene Changes in Images with Viewpoint Differences Using Graph Matching 2022

Bian et al. Unsupervised Domain Adaptation for Point Cloud Semantic Segmentation via Graph Matching 2022

Wu et al. Unsupervised Visible-Infrared Person Re-Identification via Progressive Graph Matching and Alternate Learning 2023

Liu et al. Self-supervised Learning of Visual Graph Matching 2022

Peng et al. GATE: Graph CCA for Temporal SElf-supervised Learning for Label-efficient fMRI Analysis 2022

The authors should discuss and contrast with the graph matching objectives and applications in these works to highlight their novelty in self-supervised learning.


Introduction can also be improved to better highlight the contributions. Rather than starting with discussing image-text datasets, the story should highlight the value of self-supervised learning in medical imaging and related works > proposed self-supervised learning method via graph matching > large-scale dataset collected for implementing this method > experiments on downstream tasks.


**Questions:**

Can the authors clarify how they use pre-trained ResNet-50 for downstream segmentation via U-Net? Does this mean that the encoder of U-Net has the same architecture as ResNet-50? This is not clear as the vanilla U-Net architecture (Cicek et al. 2015) is different.

**Limitations:**

Limitations are discussed.

---

> ### Author Rebuttal · Authors · 2023-08-08
>
> Thank you very much for your strongly positive feedback!
>
>
>
> **Question 1: Missing related works on graph matching in computer vision Section 2 & Improving further introduction part to highlight contributions.**
>
> ​​We sincerely acknowledge your constructive feedback. Your suggestions are valuable to us, and we will integrate these points properly to enhance both the introduction and the section on related works. These improvements certainly will make the paper more appealing, thereby highlighting our novelties.
>
>
> &nbsp;
> &nbsp;
> &nbsp;
>
> **Question 2: Clarifying how to use pre-trained ResNet-50 for downstream segmentation via U-Net.**
>
> It means that we replace the encoder network of U-Net architecture (Cicek et al. 2015) with the ResNet-50 architecture (feature outputs after five blocks of ResNet layers). The decoder layer of U-Net is then constructed so that the output at each up-sampling layer has the same feature maps as the original decoders of U-Net.
>
> In practice, using ResNet-50 or other architectures like VGG or Efficient-Net in UNet is a preferred way and can be applied in several applications such as skin lesions [1], lung segmentation [2], etc. Therefore, we want to show the benefits of LVM-Med for those study cases.
>
> [1] Nguyen, Duy MH, et al. "TATL: Task agnostic transfer learning for skin attributes detection." Medical Image Analysis, 2022 \
> [2] Cheng, Dorothy, et al. “Transfer Learning U-Net Deep Learning for Lung Ultrasound Segmentation,” Arxiv 2021

---

> > ### Comment · Reviewer_xKHb · 2023-08-14
> >
> > I have read the rebuttal and keep my original score.
> >
> > Thank you,

---

> > > ### Author Response · Authors · 2023-08-14
> > > **Thank you**
> > >
> > > Dear Reviewer,
> > >
> > > Thank you very much for reading our response and keeping the original positive score.

---

### Official Review · Reviewer_XBHj · 2023-07-05

**Soundness:** 4 excellent
**Presentation:** 4 excellent
**Contribution:** 4 excellent
**Rating:** 8
**Confidence:** 4

**Summary:**

This paper collects a large medical imaging dataset, and it also shows that a self-supervised learning technique based on second-order graph-matching enhances performance in various downstream medical imaging tasks compared to other supervised learning methods and foundation models trained on image-text instances. The evaluation also considers two different architectures: ResNet-50 and ViT backbones.

**Strengths:**

- Creating such a large medical imaging dataset is a commendable feat, and is needed by the community.
- The proposed self-supervised task is interesting.
- The evaluation results are comprehensive and thorough.

**Weaknesses:**

None to report.

**Questions:**

None.

**Limitations:**

These are included in the paper.

---

> ### Author Rebuttal · Authors · 2023-08-08
>
>
> Thank you very much for your strongly positive feedback!
>
> This encourages us to continue to improve and extend our research.

---

> > ### Comment · Reviewer_XBHj · 2023-08-22
> >
> > I have read the comments, and I keep my original score.

---

> > > ### Author Response · Authors · 2023-08-22
> > > **Thank you**
> > >
> > > Dear Reviewer,
> > >
> > > Thank you very much for reading our comments and keeping the original strongly positive score!
> > >
> > >
> > > Best,
> > >
> > > Authors

---

### Official Review · Reviewer_jRh7 · 2023-07-06

**Soundness:** 3 good
**Presentation:** 2 fair
**Contribution:** 2 fair
**Rating:** 6
**Confidence:** 4

**Summary:**

The paper proposes a set of networks called LVM-Med which are trained on large-scale medical datasets. The authors collected more than a million medical images from more than 50 publicly available datasets of diverse modalities and structures of interest (e.g. CT, MRI, Ultrasound...). In the work, several self-supervised algorithms are benchmarked on the large dataset. Furthermore, this work proposes a self-supervised contrastive learning algorithm based on a second-order graph-matching formulation.

**Strengths:**

- the paper combines an incredibly large number of medical image modalities and images

- I like the formulation of contrastive learning as a graph matching objective

- the method section is comprehensive and the contributions are formalized

**Weaknesses:**

1) Some aspects of the experimentation are unclear to me. From how I understand the text, the authors aim to compare to a large number of other datasets, baselines ("In 2D settings, we also compare with 2D supervised architectures, such as U-Net, U-Net++, Attention U-Net, etc."), and tasks across 2D and 3D. What I do not understand is how the authors choose their baselines and what they present in the tables. For example, in Table 2. for the Drive segmentation dataset, the authors report 2D supervised Methods (e.g., UNet) with Dice scores ranging from 59 to 65. Clearly, this is not a performance on par with scores reported in other works. From the literature, the state of the art in supervised DRIVE segmentation should be way higher. A fully supervised segmentation baseline on the DRIVE dataset should have 80+ Dice (https://paperswithcode.com/sota/retinal-vessel-segmentation-on-drive). Similarly, the IoU performance of BRATS baselines should be higher https://arxiv.org/pdf/1811.02629.pdf

Potentially I misunderstand what kind of comparisons the authors provide here. Can the authors please explain the choice of their baselines and their experimental settings?

2) The clarity of the writing in some sections should be improved, e.g., in the experimentation.

3) The reproducibility of the results and methods is a concern. I have not seen the code. Furthermore, the sheer amount of computing required  makes reproducibility challenging.

**Questions:**

1) The work has been run on a large set of data. Can the authors provide more detail on the overall computing required to replicate their experiments?

2) The work uses the reparameterization trick to create a complex discrete distribution. Can the authors explain the effect of the backpropagation and what this implies for the learning signal in more detail?

3) Why are there no baselines for alternative ways to define a self-supervised contrastive loss in the presented setting?

I would like to stress that I have an overall positive impression of the work and would be willing to reconsider my rating based on the rebuttal.

**Limitations:**

-

---

> ### Author Rebuttal · Authors · 2023-08-08
>
> Thank you very much for your positive and constructive feedback!
>
> **Question 1: Unclear experiment settings**
>
> **1.a: How to choose baselines?**
>
> We employ four primary baseline types (e.g., in Table 2):
>
> 1.  **2D Supervised Method**: Comparison with standard medical architectures initialized from ImageNet, while our model uses LVM-Med weights.
> 2.  **2D-SSL on Medical Data**: Benchmarking our self-supervised algorithm against state-of-the-art SSL methods trained on the same collected dataset.
> 3. **Foundation Model**: Evaluating the performance of large vision/vision-language models (e.g., Clip, Flava, SAM) compared to LVM-Med's domain-specific approach.
> 4. **Prompt-based Segmentation**:  User-interactive prompt for segmentation masks, highlighting LVM-Med's practicality besides end-to-end training as in (1), (2), and (3).
>
> **1.b: Drive and BRATS segmentation performance lower than literature**
>
> Thank you for pointing out these interesting points. After examining the papers mentioned by the Reviewer, we found out that the settings in our work are different, e.g., we use less training data (Drive) or just a single input rather than multiple inputs (BraTS).
>
> - **Drive Segmentation**: Our U-Net baseline with ResNet-50 (LVM-Med) achieved a 65 Dice score, differing from the literature (81 Dice) due to distinct settings. In particular, authors in [2]  use overlapping image patches with a stride of 32 and an image size of 128 × 128 for training. Therefore **they ended up with 4200 images for training (we used 20 images as in the original training set)**. During testing, they applied overlapping image patches again with a stride of 3 and averaged predictions over 20 sub-patches to predict for each image in the test set.
>
>    To provide further  insights, we conduct additional experiments on the Drive dataset using U-net load LVM-Med ResNet-50. Our performance given the same settings as [2] is **84.2 Dice score on average, which works better than the baseline in [2]**.
>
> - **BRATS-2018 Segmentation**:  Our 3D-IoU score is 73 for the whole tumor, compared to 88-90 Dice score by the best method [3]. There are two main different settings: (i) First, **we only use a Flair format for each patient, while [3] combined four available 3D MRI modalities of each patient into the 4-channel image as an input**. (ii) Second, we measure performance on a test set with 95 samples randomly selected from a training set of 285 patients while [3] reported performance on a test set of the BraTS-2018 competition, which is unavailable now because the competition has been closed.
>
>
> In conclusion, we utilize simple configurations for all datasets, skipping extra pre-processing for data augmentation (Drive) or input fusion (BraTS). We believe these default settings better showcase the benefits of using pre-trained models, especially with limited labeled data. Nevertheless, further experiment details will be included in the Appendix to avoid confusing readers. We appreciate the Reviewer's insightful feedback.
>
>
> [2]  RV-GAN: Segmenting Retinal Vascular Structure in Fundus Photographs using a Novel Multi-scale Generative Adversarial Network, MICCAI-2021
>
> [3] 3D MRI brain tumor segmentation using autoencoder regularization, 4th International Workshop, BrainLes 2018, MICAI.
>
> **Question 2: Reproducibility and code availability**
>
> We are committed to open-sourcing our code and pre-trained models. **We have sent the code to the Area Chair. We refer the reviewer to the Area Chair for obtaining the code**. In the code, we present detailed instructions in README.md and release ResNet-50 and ViT models trained by LVM-Med, along with the configurations of segmentation, classification, and object detection tasks.
>
>
> **Question 3: Computing details**
>
> As mentioned in **Section 4.1 Implementation details**, we trained ResNet-50 and ViT-B/16 on our dataset using high-powered GPU systems with 16 A100-GPUs, each with 80GB memory. The ResNet-50 took five days (batch size: 3200 images), and ViT-B took seven days (batch size: 2800 images) for 100 epochs.
>
> While the training procedures require a high-powered GPU system, our open-source weights will significantly reduce the time and financial investments necessary for other studies seeking to apply our results in medical applications.
>
>
> **Question 4: Reparameterization trick and backpropagation**
>
> We utilize IMLE for reparameterizing discrete distributions, outlined in Algorithm 1. To assess the efficacy of this backpropagation approach, we conducted a comparative analysis. Specifically, we compared our method (referred to as LVM-Med (Full) in Table 6) to an alternative technique employing constant value-based perturbation [50] (referred to as LVM-Med w/o Gumbel noise, Table 6). Through comprehensive evaluation across classification and segmentation tasks, we have consistently achieved superior performance than the alternative approaches [50]. For example, segmentation results with IMLE are 83.05, declining to 81.37 using [50].
>
> [50]  Optimizing rank-based metrics with blackbox differentiation, CVPR 2020.
>
> **Question 5: Baselines for contrastive loss in the presented setting**
>
> We thoroughly compare our approach with alternative contrastive losses (Twin-Barlon, Dino, SimCLR, Moco-v2, VicRegl) in Tables 2, 3, 4, and more (2D-SSL on medical data), which were trained using the same dataset as LVM-Med. Moreover, we demonstrated in Figure 1 (on the right) how the algorithms of LVM-Med can serve as a unified and extending framework for other contrastive SSL algorithms.
>
> We hope our response resolved most of your concerns, and helped you evaluate our work more positively. If you have other comments, we are happy to address them in the reviewer-author discussion period.

---

> > ### Comment · Reviewer_jRh7 · 2023-08-17
> >
> > Dear Reviewers,
> >
> > thank you for your rebuttal. The clarifications helped me a lot. The overall experimentation is impressive.
> >
> > I am still a bit puzzled by the initial choice of baselines. Given the computational requirements of your work "16 A100-GPUs, each with 80GB memory", simplifying your experimentation on the relatively fast supervised methods appears odd. If possible please also provide experiments on the BRATS dataset. Maybe even consider participating in the challenge itself if your method remains superior?
> >
> > Overall I see merit in this work and uphold my rating leaning to accept.

---

> > > ### Author Response · Authors · 2023-08-18
> > >
> > > Dear Reviewer,
> > >
> > > Thank you very much for reading our response, providing additional feedback, and upholding the positive rating.
> > >
> > > Since LVM-Med proposes novel pre-trained models, it is essential to assess their performance in downstream tasks with pure settings, i.e., avoiding adding extra pre-processing or increasing training data by data augmentation. Otherwise, it is difficult to justify whether improved performance comes from pre-trained models or the number of increased training instances. In most conducted experiments, we tried to examine this factor either with segmentation (Table 2,3) or classification (Table 5, linear evaluation and fine-tuning setting).
> > >
> > >
> > > We also validated LVM-Med performance in complex configurations, encompassing diverse factors such as architectures, incorporating supplementary training data or features, etc. For instance, Figure 3 in the main paper shows our performance in diabetic retinopathy grading tasks where we use the DRG-Net network and load our pre-trained model (ResNet-50) to this architecture. The results demonstrate our strategy leads to state-of-the-art records compared to the latest method in this benchmark.
> > >
> > >
> > > Finally, **we employ the computational resources involving 16 A100-GPUs, specifically during the pre-training phase. It is essential to note that these resources are not utilized in the downstream tasks**. For the downstream tasks, we resort to modest GPUs, e.g., a single RTX 3090 with 24GB memory, to load our model and fine-tune it. Such computation costs are equal to other baselines like U-Net, or U-Net ++ (2D Supervised Method baselines), which reasons why we compare those approaches in experiments.
> > >
> > > For the results of the BraTS dataset using the similar settings as the challenge, we are implementing this and will get back to you when the results are ready. In the meantime, if the reviewer has other questions, we are happy to discuss them.

---

> > > > ### Comment · Reviewer_jRh7 · 2023-08-21
> > > >
> > > > Thanks again, I now understand your choices and increase my rating.

---

> > > > > ### Author Response · Authors · 2023-08-21
> > > > >
> > > > > Dear Reviewer
> > > > >
> > > > > We greatly appreciate your feedback and for increasing your rating. As mentioned in the previous comment, we want to update the results on BratS 2018 dataset using LMV-Med’s weights.
> > > > >
> > > > > In short, we achieved an **average of 87.54 3D-Dice scores, while the SOTA is approximately 87.14 across three trial times**. The setting is the following:
> > > > >
> > > > > **Our**: We use TransUnet with LVM-Med’s ViT architecture. We also combine all 3D volumes as SOTA [3] with four available 3D MRI modalities of each patient into the 4-channel image as an input. Because [3] has another reconstruction decoder layer, we integrate a similar architecture to TransUnet.
> > > > >
> > > > > **SOTA**: We use architecture released by [3] and fine-tune on the same training/validation/testing partition as our configurations.
> > > > >
> > > > > In summary, we have found that under similar extra-input data and modified architecture, LVM-Med can perform well as state-of-the-art approaches.
> > > > >
> > > > > [3] 3D MRI brain tumor segmentation using autoencoder regularization, 4th International Workshop, BrainLes 2018, MICAI 2018

---

### Official Review · Reviewer_WEtP · 2023-07-06

**Soundness:** 3 good
**Presentation:** 3 good
**Contribution:** 3 good
**Rating:** 6
**Confidence:** 3

**Summary:**

The paper presents a large scale medical imaging dataset consisting of 1.3M medical images from 55 publicly available datasets along with a new contrastive learning framework based on graph matching. Specifically, the model is firstly pre-trained on the collected dataset, and then finetuned towards different downstream tasks, improvement is observed a cross different datasets and settings.

**Strengths:**

+ A large scale *medical* dataset for pre-training purposes

+ Well-written and easy to follow

+ Extensive experiments and good results

**Weaknesses:**

- The paper did not provide enough details regarding the proposed datasets, especially in terms of the usage of the dataset, which, in my humble opinion, is pretty important to provide guidance on training medical models on large-scale datasets, e.g., 1. how are 3D data used, are they sliced into 2D data firstly? 2. The data comes from different datasets and maybe in different modalities, any balancing strategy during pre-training? 3. Any augmentations besides multi-crop? (e.g., flip, rotate, color jittering etc). In short, I expect the author provide more details regarding how they utilize the dataset

- I appreciate the author provided dataset statistics details in the supplementary, yet I am curious how are these datasets selected and have the author tried any filtering/curation? As data curation has been considered very important in modern large-scale model training [1], it would be great to have some insights in the medical area.


[1] DataComp: In search of the next generation of multimodal datasets


**Questions:**

see weakness

**Limitations:**

see weakness

---

> ### Author Rebuttal · Authors · 2023-08-08
>
>
>
> Thank you very much for your positive and constructive feedback!
>
>
> **Question 1: Provide more details regarding the usage of the datasets.**
>
> **1.a:  How are 3D data used, are they sliced into 2D data first?**
>
> Yes, for 3D volume data, we slice them into 2D images first. We mentioned this in **Section E. Dataset overviews** in the Appendix and will emphasize it more clearly in the revised version.
>
> &nbsp;
> &nbsp;
>
> **1.b: The data comes from different datasets and maybe in different modalities; any balancing strategy during pre-training?**
>
> We did a statistical summary of the collected datasets (in Figure 5, Appendix), and the results indicate an imbalance between data modalities. To address this imbalance, we used the following strategies.
>
> 1. In each mini-batch during pretraining, we randomly select a subset of available modalities, e.g., color image, X-ray, and MRI.
>
>
> 2. To balance samples in each modality, we combine over-sampling and data augmentation to increase the total samples. Specifically, new samples from minority classes are generated by duplicating images and applying random crop operations covering 85-95% of image regions and then rescaling them to the original resolutions. Note that these augmentations are not used in the self-supervised algorithm  (operations s, t ~ T) to avoid generating identical distorted versions in this sampling procedure.
>
>      We will further discuss this sampling procedure in Section E. Dataset overviews.
>
> &nbsp;
> &nbsp;
>
> **1.c: Any augmentations besides multi-crop? (e.g., flip, rotate, color jittering, etc.)**
>
> For the augmentations, we mainly follow prior work [29] that emphasizes the importance of multi-crop augmentation. Though, in the implementation, the method is combined with other widely used operations. In particular, what we use is multi-crop $\rightarrow$ flip (probability 50%) $\rightarrow$  color jilter $\rightarrow$  random Gaussian blur $\rightarrow$  normalization.
>
> We will add more details in *Section 4.1 Implementation details*
>
> [29] Vicregl: Self-supervised learning of local visual features
>
> &nbsp;
> &nbsp;
>
> **Question 2: How are these datasets selected? Is any filtering/curation used?**
>
> In this work, we focus on building a **large-scale model for medical images which only uses raw data without any supervised signals** (e.g., segmentation masks or classification labels). This is essentially different from other multi-modal data where image-text pairs are used as inputs for the algorithm, necessitating careful data curation to prevent skewed or inaccurate information. With this characteristic in mind, we employ specific criteria to pick datasets, outlined as follows:
>
>
> 1. Collecting **as many public datasets as possible** whose number of samples is not too small (at least 100 images).
> 2. The selected dataset represents **diverse data modalities** (including X-ray, CT, MRI, color images, ultrasound, etc).
> 3. The selected data represents **diverse body organs** such as lungs, cells, brains, breasts, etc.
>
> To define a set of potential datasets, we carefully survey from different sources such as competition challenges (e.g., grand-challenge),  papers,  or benchmarks in medical-related conferences/journals. The collected datasets are then sampled with 20% images if training/testing splittings are unavailable; otherwise, we use all samples in the training set to avoid potential testing data leaking, as discussed in the paper.
>
> We hope our response resolved most of your concerns, and helped you evaluate our work more positively. If you have other comments, we are happy to address them in the reviewer-author discussion period.

---

> > ### Comment · Reviewer_WEtP · 2023-08-17
> >
> > Thanks for providing the rebuttal, most of my concerns are addressed. I appreciate the author's efforts in exploring large-scale dataset for medical imaging analysis. I increase my rating to weak accept.

---

> > > ### Author Response · Authors · 2023-08-18
> > >
> > > Dear Reviewer,
> > >
> > > Thank you very much for reading our response and increasing the rating.
> > >
> > > We will incorporate your valuable suggestions into our next revision.

---

### Author Rebuttal · Authors · 2023-08-08



We would like to thank all reviewers for the positive and constructive feedback, which we will leverage to improve this work.

We are very encouraged that most reviewers agree that our efforts in **creating such a large medical imaging dataset is important and needed by the community**. Reviewers jRh7, xKHb, and XBHj also appreciate the **LVM-Med algorithms and believe they are novel and interesting**. In addition, Reviewers WEtP, XBHj, xKHb, and yfed acknowledge the **comprehensive and thorough experiments**, which show that LVM-Med consistently **outperforms several state-of-the-art self-supervised algorithms and foundation models**.

There are a few shared concerns among Reviewers WEtP and jRh7, such as missing details for some of the experiments and improving the introduction by discussing additional relevant work (Reviewer yfed).

We appreciate your feedback and will take into account these points to improve the manuscript. For instance, missing information will be added to **Section 4.1 Implementation Details - Downstream Tasks** (main paper) and the corresponding sections in the **Appendix**. A part of the Introduction and Related Work sections will be revised based on the references provided by Reviewer yfed.

Below, we address specific concerns raised by individual reviewers in detail.

---

### Decision · Program_Chairs · 2023-09-21

**Decision:**

Accept (poster)

**Comment:**

This paper collects a large data set of publicly available medical images and conducts a comprehensive study based on graph matching techniques. Additional fine-tuning for specific tasks demonstrated significant improvements in performance for multiple scenarios. The first main contribution of the paper is the novel application of graph-matching methods. The second main contribution is an extensive set of experiments that demonstrate potentially wide applicability of the method.

The reviewers praised the abundance of experiments, the thorough approach and the novelty of the graph-matching methods for contrastive learning. The reviewer agreed that the results of this paper will benefit the community. During the discussion many remaining concerns of the reviewers  were addressed. This paper will advance our understanding of machine learning methods for medical imaging. Thus, we recommend acceptance.